# Drivers of late Holocene ice core chemistry in Dronning Maud Land: The context for the ISOL-ICE project

V. Holly L. Winton[1,2], Robert Mulvaney[1], Joel Savarino[3], Kyle R. Clem[4], Markus M. Frey[1]

[1]British Antarctica Survey, Cambridge, CB3 0ET, United Kingdom
[2]Antarctic Research Centre, Victoria University of Wellington, Wellington 6102, New Zealand
[3]University of Grenoble Alpes, CNRS, IRD, Grenoble INP, INRAE, IGE, F-38000 Grenoble, France
[4]School of Geography, Environment and Earth Sciences, Victoria University of Wellington, Wellington 6102, New Zealand

*Correspondence to*: V. Holly L. Winton (holly.winton@vuw.ac.nz) and Markus M. Frey (maey@bas.ac.uk)

**Abstract.** Within the framework of the Isotopic Constraints on Past Ozone Layer in Polar Ice (ISOL-ICE) project, we present
initial ice core results from the new ISOL-ICE ice core covering the last millennium from the high-elevation Dronning Maud Land (DML), and discuss the implications for interpreting the stable isotopic composition of nitrogen in ice core nitrate ($\delta^{15}N(NO_3^-)$) as a surface ultra-violet radiation (UV) and total column ozone (TCO) proxy. In a quest of deduced TCO for $\delta^{15}N(NO_3^-)$, an understanding of past snow accumulation changes, as well as aerosol source regions and present-day drivers of their variability are required. We therefore report here the ice core age-depth model, the snow accumulation and ice
chemistry records, and correlation analysis of these records with climate variables over the observational era (1979-2016). The ISOL-ICE ice core covers the last 1349 years from 668 to 2017 C.E. ± 3 years extending previous ice core records from the region by two decades towards the present and shows excellent reproducibility with those records. The extended ISOL-ICE record of last two decades showed a continuation of the methane sulphonate ($MSA^-$) increase from ~1800 to present while there were less frequent large deposition events of sea salts relative to the last millennium. While our chemical data do not
allow to distinguish the ultimate (sea ice or the open ocean) source of sea salt aerosols in DML winter aerosol, our correlation analysis clearly suggests that it is mainly the variability in atmospheric transport and not the sea ice extent, that explains the interannual variability in sea salt concentrations in DML. Correlation of the snow accumulation record with climate variables over the observational era showed that precipitation at ISOL-ICE is predominately derived from the South Atlantic with onshore winds delivering marine air masses to the site. The snow accumulation rate was stable over the last millennium with
no notable trends over the last two decades relative to the last millennium. Interannual variability in the accumulation record, ranging between 2 and 20 cm $a^{-1}$ (w.e.), would influence the ice core $\delta^{15}N(NO_3^-)$ record. The mean snow accumulation rate of 6.5 ± 2.4 cm $a^{-1}$ (w.e.) falls within the range suitable for reconstructing surface mass balance from ice core $\delta^{15}N(NO_3^-)$ highlighting that the ISOL-ICE ice core $\delta^{15}N(NO_3^-)$ can be used to reconstruct either the surface mass balance or surface UV if the ice core $\delta^{15}N(NO_3^-)$ is corrected for the snow accumulation influence and thereby leaving the UV imprint in the $\delta^{15}N(NO_3^-$
) ice core record to quantify natural ozone variability.

# 1 Introduction

The stratospheric ozone $O_3$ layer shields all land-based life forms from harmful ultraviolet (UV-B) radiation. Assessing the natural variability of total column ozone (TCO) before the instrumental era is critical to put modern observations (Brönnimann et al., 2003; Bodeker et al., 2021) into historical context but this has proven difficult owing to the lack of a well-constrained proxy. To address the lack of suitable UV radiation proxies, the Isotopic Constraints on Past Ozone Layer in Polar Ice (ISOL-ICE) project drilled a new 120 m ice core from the East Antarctic Plateau in 2017 to apply a new ice core proxy based on the fractionation observed in stable nitrogen ($\delta^{15}N$) and oxygen ($\Delta^{17}O$) isotopes of nitrate ($NO_3^-$) and to develop the first millennial-timescale reconstruction of past surface UV-radiation (TCO) and atmospheric oxidation processes. Interpretation of the nitrate stable isotope record ($\delta^{15}N$ and $\Delta^{17}O$) in terms of past UV (TCO) and atmospheric oxidation processes requires a thorough understanding of the local air-transfer function and its driving parameters. The stable isotope of nitrogen $^{15}N$ in $NO_3^-$ ($\delta^{15}N(NO_3^-)$) in surface snow is highly enriched compared to atmospheric $NO_3^-$ (e.g., Erbland et al., 2013; Cao et al., 2022; Winton et al., 2020) due to $NO_3^-$ loss and redistribution from snow, which is driven by UV-photolysis (Frey et al., 2009; Berhanu et al., 2014; Shi et al., 2019). $NO_3^-$ isotopic fractionation is strongest at sites with very low snow accumulation rates (2-3 cm a$^{-1}$ (w.e.), such as Dome A and Dome C on the East Antarctic Plateau, due to enhanced $NO_3^-$ post-depositional recycling which erases the source signature of $\delta^{15}N(NO_3^-)$ due to longer exposure of surface snow layers to incoming UV radiation before burial (Shi et al., 2022a; Frey et al., 2009). Higher rates of snow accumulation rate to 6 cm a$^{-1}$ (w.e.) and above is sufficient to preserve the seasonal cycle of $\delta^{15}N(NO_3^-)$ in the snowpack (Winton et al., 2020) and enrichments in $\delta^{15}N(NO_3^-)$ by the effects of snow accumulation superimpose those due to stratospheric ozone depletion (Cao et al., 2022; Shi et al., 2022b). The inverse relationship between the snow accumulation rate and $\delta^{15}N(NO_3^-)$ is well constrained across Antarctica so that for ice core sites within the transition zones between the dome summits and the coast with a snow accumulation rate between 4 and 20 cm a$^{-1}$ (w.e.), ice core $\delta^{15}N(NO_3^-)$ has been proposed as proxy for surface mass balance (Akers et al., 2022).

As a first part of the ISOL-ICE project, an air-snow transfer study was carried out at the ice core drill site (Winton et al., 2020). At the ISOL-ICE ice core site located on the high East Antarctic Plateau at Kohnen Station (75°00'06"S, 0°04'04"E, 2892 m amsl) in Dronning Maud Land (DML), the sensitivity of $\delta^{15}N(NO_3^-)$ to changes in snow accumulation (and seasonality), TCO and snow e-folding depth (light attenuation) was assessed using atmospheric, surface snow and snow pit observations of $\delta^{15}N(NO_3^-)$ supported by air-snow transfer modelling (Winton et al., 2020). The dominant factors controlling the archived $\delta^{15}N(NO_3^-)$ signature at the ISOL-ICE site are the e-folding depth and snow accumulation rate, followed by TCO. Taking past changes in snow accumulation into account, it may be possible to reconstruct past UV and TCO on longer timescales from the $\delta^{15}N(NO_3^-)$ signal in the ISOL-ICE ice cores, provided other factors such as the e-folding depth have remained the same within the target resolution time. Thus, in order to interpret the ice core $\delta^{15}N(NO_3^-)$ record, the accumulation rate needs to be independently derived from annual layer counting and volcanic horizons to account for post-depositional isotope fractionation not related to surface-UV changes.

Furthermore, constraining the aerosol source regions for the ISOL-ICE ice core has implications for interpreting the ice core record of $NO_3^-$ and its stable isotopes. The net primary sources of $NO_3^-$ to the Antarctic atmosphere and snowpack are methyl nitrate ($CH_3NO_3$) from open ocean and sea ice, and stratospheric $NO_3^-$ (Burger et al., 2022; Burger et al., 2023; Mulvaney and Wolff, 1993; Savarino et al., 2007; Jacobi et al., 2000; Jones et al., 1999; Beyersdorf et al., 2010). Ice core proxies of open ocean and sea ice aerosol include biogenic sulfur species and sea salts (Frey et al., 2020; Abram et al., 2013; Thomas et al.,
2019; Thomas et al., 2023). At some ice core sites, atmospheric transport strength, rather than aerosol source, is the dominant factor that determines the chemical signal preserved in ice cores. For example, methanesulfonic acid (MSA) is derived from the oxidation of dimethylsulfide (DMS, $(CH_3)_2S$) from marine phytoplankton emissions to the atmosphere, and reflects atmospheric transport rather than sea ice conditions in coastal Weddell Sea and Dronning Maud Land ice cores (e.g. Fundel et al., 2006; Abram et al., 2007). Thus, evaluating the ISOL-ICE ice chemistry derived from oceanic sources is necessary to
understand the relationships between ice core $NO_3^-$ and its sources.

As a second step to developing the UV (TCO) ice core proxy within the framework of the ISOL-ICE project, we present the full ice core record of annual snow accumulation and ice core chemistry, and the age-depth model. The ISOL-ICE ice core extends the European Program for Ice Coring in Antarctica (EPICA) Dronning Maud Land (EDML) record by two decades
towards the present. We discuss the aerosol source regions and links to climate variability to provide context for potential $NO_3^-$ sources. The development of the site-specific accumulation record provides foundational data to model the accumulation component of the $\delta^{15}N(NO_3^-)$ record. These initial ice core results, together with the ISOL-ICE air-snow transfer study (Winton et al., 2020) set the framework for the interpretation of the 1000-year ice core record of stable nitrogen and oxygen isotopes from the high-elevation DML. The final step in developing the millennial-timescale reconstruction of past surface UV-
radiation (TCO) is to use the ice core $NO_3^-$ concentration and accumulation records reported in this study, along with the measured ISOL-ICE ice core $\delta^{15}N(NO_3^-)$, to model the site-specific surface conditions using the inverted TRANSITS model (Jiang et al., 2023). The UV (TCO) proxy is extracted from the $\delta^{15}N(NO_3^-)$ record by constraining the accumulation rate effect on the $\delta^{15}N(NO_3^-)$ thereby leaving the UV effect. Application of the UV (TCO) transfer function to the ice core $\delta^{15}N(NO_3^-)$ will be presented in a future study.

## 2 Methods

### 2.1 Ice core drilling

A new ice core was drilled as part of the ISOL-ICE project at Kohnen Station, located in DML on the high-elevation East Antarctic Plateau (2892 m a.s.l) approximately 550 km from the coast. The ISOL-ICE ice core site was located within the clean air sector ~1 km from the deep EDML core, which was drilled in 2001-2006 to a depth of 2761 m (Wilhelms et al., 2017). The 120 m deep ISOL-ICE ice core (core A; 74.9961 °S, 0.09472 °E; Figure 1) was drilled in January 2017 using the British Antarctic Survey (BAS) shallow electromechanical ice core drill (Mulvaney et al., 2002), which produces a core with a 105 mm internal diameter in a dry borehole. Two additional 10 m firn cores (cores B and C) were recovered from the site using a stainless-steel drill with an internal diameter of 96 mm. A fourth 2 m firn core (core CF) was recovered using a carbon fiber tube with an internal diameter of 98 mm. The core sections were cut into 80 cm lengths and shipped frozen to the BAS in Cambridge, United Kingdom and stored at -25 °C until core processing. In addition, two snow pits were sampled for soluble ions and nitrate stable isotopes. Core C was analysed for perfluorocarboxylate. Results from the snow pits and firn core C can be found in Winton et al. (2020) and Garnett et al. (Garnett et al., 2022) respectively.

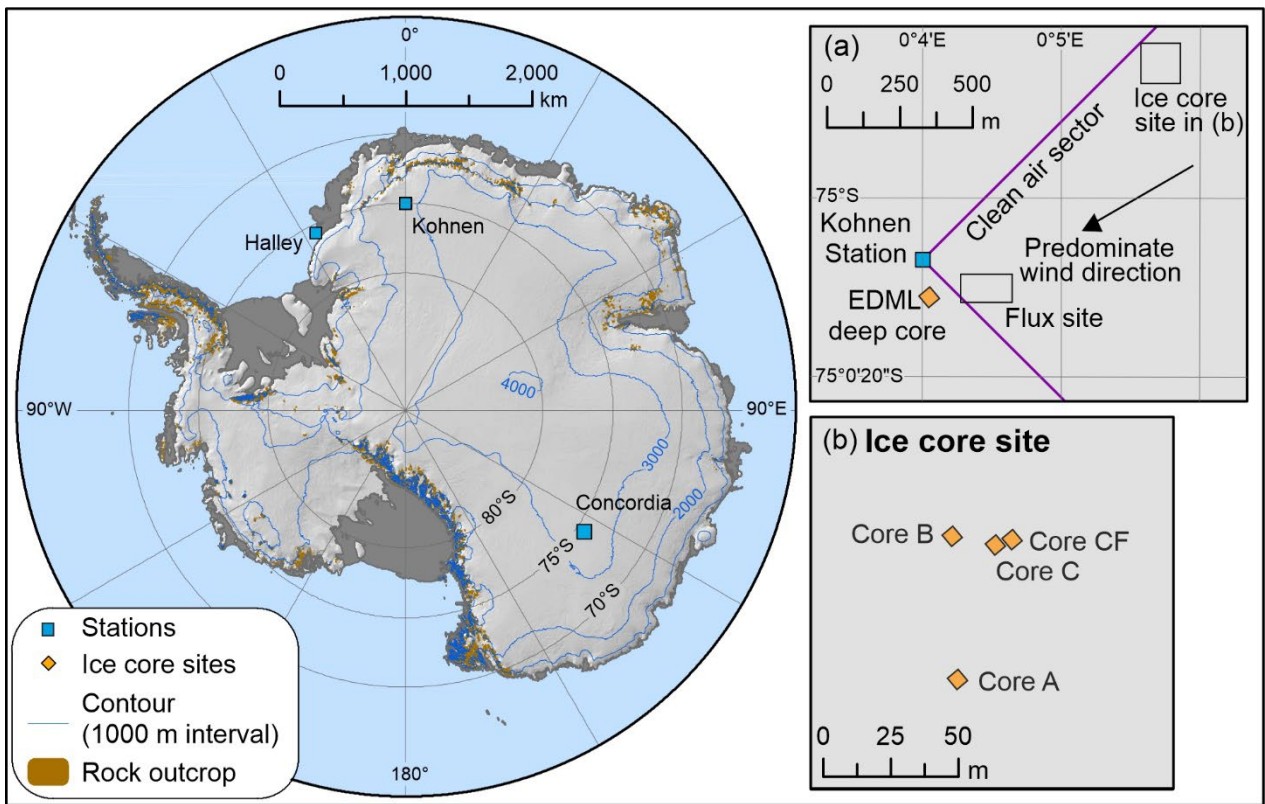

**Figure 1: Map of the ISOL-ICE ice core drilling site in the clean air sector of Kohnen Station. (a) Insert of Kohnen Station in Dronning Maud Land (DML) highlighting the predominate wind direction, deep EDML ice core site, and the ISOL-ICE "ice core"**

and "flux" sites. The latter is described in Winton et al. (2020). (b) ISOL-ICE "ice core site" showing the main ice core (core A) and shallow firn core (cores B, C and CF) locations. Modified from Winton et al. (2020).

## 2.2 Ice core processing and glacio-chemical analysis

At the BAS, the main ice core (core A) was logged, analysed by dielectric profiling (DEP), and then cut into sections for continuous flow analysis (CFA), analysis of nitrate stable isotopic composition, and archival (Figure A1). Firn cores B and CF were also logged and analysed by DEP. Here we present initial results from the CFA and DEP analysis as a basis for interpreting the nitrate concentration, stable nitrate isotope, and insoluble particle records which will be presented in subsequent studies. The ISOL-ICE ice chemistry data are available through the Polar Data Centre (Winton et al., 2019).

Dielectric conductivity of the full ice core was determined by DEP (Oerter et al., 2000; Moore and Paren, 1987; Wilhelms et al., 1998; Hofstede et al., 2004). The DEP data were taken at 250 kHz in 5 mm increments with a 10 mm long measuring electrode. Cores were measured only after temperature equilibration in the cold laboratory to reduce the effects of temperature-dependent activation energy on the DEP and corrected for minor differences in core temperature measured by a probe inserted between the electrodes and the ice core. Repeat measurements were made on the main core A between 35 and 39 m and showed excellent reproducibility ($R^2$=0.93, $P$ <0.001; RMSE=0.28 µS cm$^{-1}$). Mean background differences between the DEP values in the 3 cores also reflect the different core diameters since the dielectric response measured by the DEP is proportional to the volume of ice between the measurement plates; this is evident around 40 m depth in Figure 2c where several meters of ice were cut with the core cutters set to a smaller diameter. The rising trend observed in Figure 2c reflects the increasing density of the ice with depth.

A stainless-steel bandsaw was used to cut a 32 x 32 mm ice stick for CFA of trace elements, soluble anions, hydrogen peroxide ($H_2O_2$), insoluble particles and liquid conductivity. CFA sticks were melted vertically on a heated, gold-plated copper melt head, which consists of two rings separating the inner and outer section of the meltwater. The inner section of the CFA meltwater was used for chemical analysis ensuring no contact of the meltwater with exposed ice core surfaces. The CFA system used was an earlier version of that described by Grieman et al. (2022).

Soluble anions were measured using fast ion chromatography (FIC; ICS-3000, Dionex™). A suite of anions, including $NO_3^-$, chloride ($Cl^-$), methane sulphonate ($MSA^-$; $CH_3SO_3^-$) and sulfate ($SO_4^{2-}$), were determined using an AS15-5 µm and AERS500-4 mm suppressor. The sampling resolution of the FIC measurements was 5 cm at the top of the core and 4 cm at the bottom resulting in an average of 2 ± 1 measurements per year. A certified reference material (CRM; European Reference Material ERM-CA408 simulated rainwater) containing $NO_3^-$, $SO_4^{2-}$ and $Cl^-$ was measured at the beginning, end and during each laboratory analysis day to determine the accuracy and precision of the soluble anion measurements which are reported in Table A1.

High-resolution measurement of trace elements, including sodium (Na) and magnesium (Mg), were made using inductively coupled plasma-mass spectrometry (ICP-MS; 7700 series, Agilent). A CRM (European Reference Material, ERM-CA616 groundwater) containing Na and Mg was measured at the beginning, end and during the course of each day to determine the accuracy and precision of the trace element measurements which are reported in Table A1. Hereafter, Na and Mg measured by ICP-MS are assumed to represent their ionic form $Na^+$ and $Mg^{2+}$ (Grieman et al., 2022).

Liquid conductivity was measured in the CFA fluid line using a digital conductivity meter (Dionex™ 2010i) with a flow through cell (Amber Science, Model 1056).

Insoluble particle counts were performed using a laser particle detector (Abakus, Fa.Klotz, Bad Liebenzell, Germany) which was connected to the meltwater flow with a flow rate of 1.6 mL min$^{-1}$. The instrument was set to measure insoluble particles in 32 individual diameter size bins ranging between 0.9 and 50 μm. The dust mass was calculated from the measured volume assuming a spherical particle density of 2.5 g cm$^{-3}$. The sampling resolution of the ICP-MS, liquid conductivity and insoluble particle measurements was <1 mm resulting in an average of $190 \pm 120$ measurements per year in the top 5 m of the core and an average of $140 \pm 60$ measurements per year in the bottom 5 m of the core.

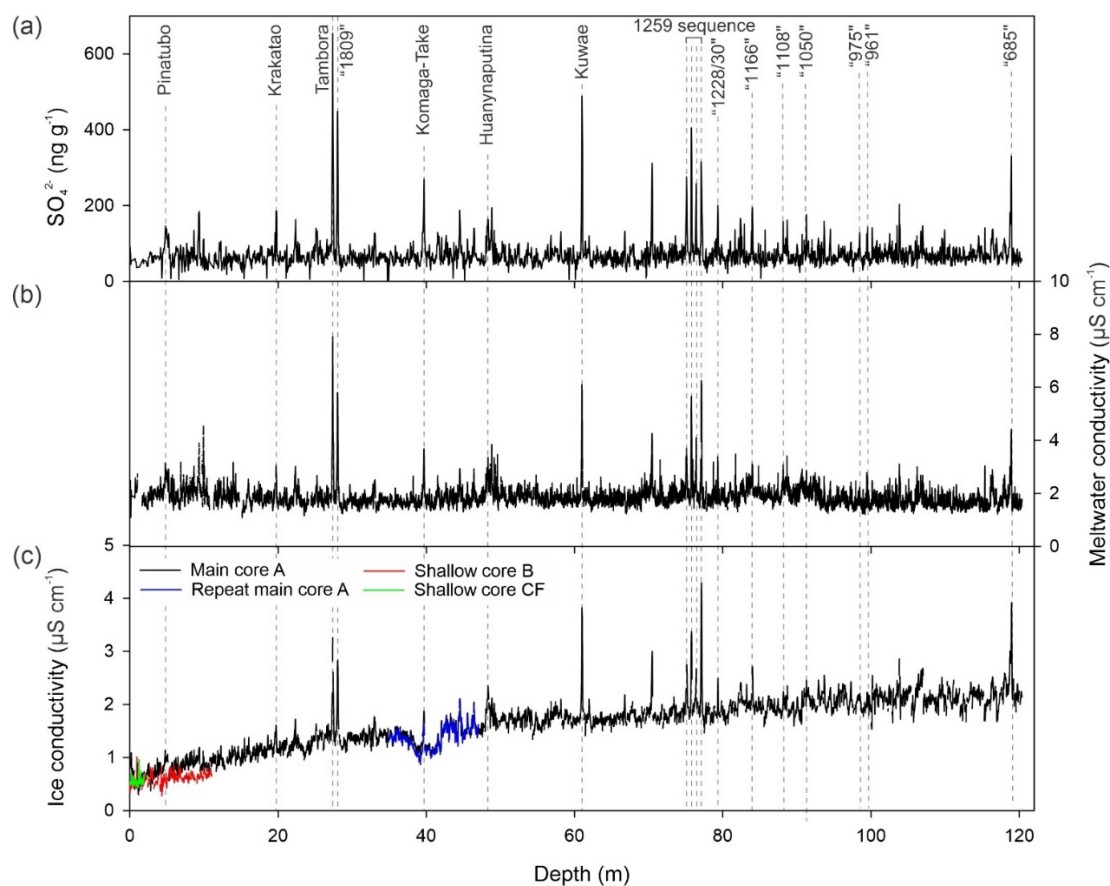

**Figure 2. Volcanic markers identified in the ISOL-ICE core using (a) sulfate (SO$_4$$^{2-}$)_concentrations, (b) liquid conductivity, and (c) ice conductivity (dielectric profile; DEP). The unknown volcanic eruptions are documented in the literature for ice cores in the Dronning Maud Land region. DEP data were corrected for temperature but not density. Note the diameter of the core reduced to 98 mm between 38 and 42 m which is reflected in the lower DEP values in the repeated section of the main core A.**

## 2.3. Ice core dating

The age-depth model of the ISOL-ICE ice core was constructed using annual layer counting constrained by well-dated volcanic eruption horizons documented in the literature based on ice cores in the Dronning Maud Land region (e.g. Sigl et al., 2022). Annual layer counting was based on the high-resolution Na$^+$ and Mg$^{2+}$ concentrations following previous ice core and aerosol studies in DML (Göktas et al., 2002; Garnett et al., 2022; Ming et al., 2020). Multi-year aerosol observations from Kohnen Station exhibit a narrow seasonal minimum in Na$^+$ and Mg$^{2+}$ concentrations during austral summer and a broad peak in late winter/spring (Weller and Wagenbach, 2007). Annual markers (January) were positioned at the sharp, well defined summer Na$^+$ and Mg$^{2+}$ minima. Volcanic tie points were identified using SO$_4$$^{2-}$ concentrations, liquid conductivity and DEP and were placed at the depths reported in Table A2 and illustrated in Figure 2. The volcanic dates were taken from either the eruption date if known or published deposition dates used to construct age-depth models for ice cores in the DML region if unknown (e.g. Cole-Dai and Mosley-Thompson, 1999; Zielinski et al., 1994; Langway et al., 1995; Traufetter et al., 2004).

## 2.4. Snow accumulation rate and density

The density of each 80 cm ice core bag was calculated using the ice core mass, diameter and length. Snow accumulation rates were derived from the density measurements and annual layer thickness of the core. Plastic deformation associated with thinning of deeper ice layers was accounted for by using the thinning model of Nye (1963) which resulted in a 10 % correction of the accumulation rate at the base of the core.

## 2.5. Relationships between accumulation rate, glacio-chemistry and large-scale climate variability

To understand the transport mechanisms and source regions of moisture, snowfall and aerosols to the ISOL-ICE site, the snow accumulation and glacio-chemistry ice core records were correlated with climate variables over the observational era (1979-2016). Correlations were performed on annual averaged data (January-December), following Thomas et al. (2023), to correspond with the annually-resolved accumulation and MSA$^-$ records, and seasonally averaged data resulting from the sub-annually-resolved Na$^+$ and Mg$^{2+}$ measurements. Seasonal means were grouped into austral summer-autumn (December to May average) and austral winter-spring (June to November average) for each year using linear interpolation between annual markers. Despite extreme Antarctic precipitation events and nonlinear snowfall throughout a year (Turner et al., 2019), the climatology of monthly mean Na$^+$ and Mg$^{2+}$ concentrations exhibits the same seasonal trend as aerosol observations at Kohnen Station, i.e., a narrow seasonal minimum in austral summer and a broad peak in late winter/spring (Weller and Wagenbach, 2007). These seasonal grouping reflects the known aerosol Na$^+$ concentration peak in late winter/spring (Weller and Wagenbach, 2007), the distinct ice core Na$^+$ concentration peak between June and September, the relatively coherent seasonal Na$^+$ and Mg$^{2+}$ concentration patterns and background climate patterns, e.g., the winter transition and sea ice advance season and the summer transition and sea ice retreat season, respectively. Correlations with NO$_3^-$ concentrations were not calculated due to missing NO$_3^-$ concentration data in the top 8 m of the record corresponding to the observational era.

The records were correlated with select climate variables from ERA5, the fifth generation European Centre for Medium Range Weather Forecast (ECMWF) atmospheric reanalysis data (Hersbach et al., 2020), during the modern satellite observing period 1979-2016. The variables analysed include mean sea level pressure (MSLP), 10 m meridional wind (V10M), 500-hPa geopotential height (Z500) and 500-hPa meridional winds (V500) to investigate regional and large-scale circulation patterns associated with air mass and aerosol transport. The 10 m meridional wind level was chosen to investigate regional scale circulation patterns associated with surface air mass and marine aerosol transport. The 500 hPa level was chosen to capture broader-scale circulation and is reflective of the near-surface winds over the higher elevations of the Antarctic continent, such as at the high-elevation ISOL-ICE site which has an average surface pressure of 675 hPa between 1997 and 2017 (Utrecht University Automatic Weather Station (AWS) at DML05/Kohnen (AWS9; https://www.projects.science.uu.nl/iceclimate/aws/files_oper/oper_20632, last access: 29 March 2017), but is less influenced by localised near-surface boundary layer features such as katabatic winds. The records were also correlated with sea ice

concentration from the Met Office Hadley Centre Sea Ice and Sea Surface Temperature data set (Rayner et al., 2003; Titchner and Rayner, 2014) to investigate relationships between atmospheric transport pathways to the site and areas of sea ice anomalies. Lastly, the ISOL-ICE ice core records were correlated with several large-scale climate mode indices, including the observation-based Southern Annular Mode (SAM) index (Marshall, 2003), and for the El Niño-Southern Oscillation (ENSO) we employ the Niño 3.4 index and Southern Oscillation Index (SOI). The SAM index reflects the difference in mean sea level pressure anomalies between the southern middle latitudes and Antarctica, whereby positive (negative) SAM index values reflect positive (negative) pressure anomalies in middle latitudes, negative (positive) pressure anomalies over Antarctica, and a stronger/poleward (weaker/equatorward) mid-latitude jet. The Niño 3.4 index is the sea surface temperature anomaly in the east-central equatorial Pacific area-averaged over the region 5°N-5°S, 170-120°W, with positive (negative) values associated with El Niño (La Niña) conditions. The SOI reflects the atmospheric component of ENSO, and is the difference in mean sea level pressure anomalies between Tahiti (central tropical Pacific) and Darwin, Australia (west Pacific warm pool), whereby positive (negative) SOI values reflect La Niña (El Niño) conditions. The SAM index was accessed freely online at http://www.nerc-bas.ac.uk/icd/gjma/sam.html, and the Niño 3.4 index and SOI are from the Climate Prediction Center and were accessed freely online at https://www.cpc.ncep.noaa.gov/data/indices (all last accessed 22 July 2022). Annual averages of the SAM, were calculated from the monthly-mean values and are based on the standard 12-month calendar year (January-December), while seasonal boundaries (June to November and January to April) reflecting the temporal cycle of ENSO were used for Niño 3.4 and SOI indices rather than a calendar year which would split an ENSO event as May is generally the transition month between ENSO cycles (e.g. Crockart et al., 2021). The SAM index begins in 1957, and the correlations were calculated over the common period of 1957-2016 along with the reanalysis period of 1979-2016.

## 3. Results

### 3.1. Age-depth model

As with previous ice core dating studies in the high-elevation DML region, the ISOL-ICE ice core displayed clear seasonality and distinct summer $Na^+$ and $Mg^{2+}$ troughs throughout the record enabling annual layer counting for the full 120 m record. The high-resolution $Na^+$ and $Mg^{2+}$ annual signals were evaluated simultaneously. The lower resolution soluble anion measurements by FIC did not resolve seasonal signals capable for annual layer counting. The $H_2O_2$ data was also evaluated for summer peaks or winter minima, however the seasonal signal was smoothed below 2 m depth and was not suitable for annual layer counting at the low accumulation rate site. Nineteen volcanic time markers were identified from the $SO_4^{2-}$ concentrations, liquid conductivity and DEP. All volcanic horizons were well aligned in the three records (Figure 2). The difference between the conductivity measurements of the ice (DEP) and meltwater (liquid conductivity), i.e., the increasing trend in the firn section of the DEP record, reflects the change in density with depth compared to the meltwater conductivity measurements where there is no density effect. The conductivity values from the ice and meltwater measurements agree well below ~75 m depth. Annual layer counting of $Na^+$ and $Mg^{2+}$ concentrations constrained by the volcanic horizons indicates that

the ISOL-ICE ice core spans the last 1349 years from 2017-668 C.E. extending the EDML core by two decades towards the present (Ruth et al., 2007). A maximum age uncertainty in our layer counting of $\pm$ 3 years is estimated between 48-61 m corresponding to about 150 yr (1453-1601 AD), which is one of the largest intervals enclosed between documented volcanic peaks in Antarctic ice cores (Table A2).

## 3.2. Snow accumulation record

The ISOL-ICE snow accumulation record is presented Figure 3a. The mean accumulation rate for the entire core is 6.5 $\pm$ 2.4 cm a$^{-1}$ (w.e.) consistent with previous estimates of 6.0–7.1 cm a$^{-1}$ (w.e.) from ice cores from the high-elevation DML region (Sommer et al., 2000; Oerter et al., 2000; Hofstede et al., 2004). The mean of the 1995-2017 extension of snow accumulation record is 6.0 $\pm$ 2.7 cm a$^{-1}$ (w.e.) indicating that the accumulation rate has been relatively stable over the last millennium. While the record does not display centennial scale variability in the snow accumulation rate, there is significant interannual variability in the record ranging between 2 and 20 cm a$^{-1}$ (w.e.).

## 3.4. Ice chemistry

The ISOL-ICE ice chemistry records are displayed in Figure 3. $Na^+$, $Mg^{2+}$ and $Cl^-$ in Antarctic ice cores are derived from marine sea salt either originating from the open ocean or from sea ice sources such as blowing snow or frost flowers (e.g. Abram et al., 2013). The concentrations of $Na^+$, $Mg^{2+}$ and $Cl^-$ marine aerosols have been stable over the last millennium with mean concentration values of 25.4 $\pm$ 18.9, 2.5 $\pm$ 2.1 and 39.0 $\pm$ 23.8 ng g$^{-1}$ respectively. The extended ISOL-ICE record showed that there were less frequent sea salt deposition events over the most recent two decades relative to the last millennium. Sulfate deposited to ice cores is derived from mixed sources including marine primary production, sea salt and volcanic eruptions (e.g. Sigl et al., 2014; Lin et al., 2022). Sulfate concentrations, with a mean value of 70.6 $\pm$ 37.2 ng g$^{-1}$, also did not display shift in the baseline. The ISOL-ICE $SO_4^{2-}$ record shows excellent reproducibility with other DML ice cores (Traufetter et al., 2004; Sigl et al., 2022) where the large $SO_4^{2-}$ peaks are attributed to volcanic sources (Figure 2). The sole source of $MSA^-$ is from the oxidation of dimethylsulfide (DMS, $(CH_3)_2S$) from marine phytoplankton emissions to the atmosphere. While sea salt deposition was relatively constant over the last millennium, $MSA^-$ and $NO_3^-$ were the only analytes that displayed centennial scale variability with a gradual decrease in concentrations from 668 to 900, stable concentrations from 900 to 1500 and a notable increase from 1800 to present for $MSA^-$ and from 1500 to present for $NO_3^-$. The centennial scale temporal variability of $MSA^-$ and $NO_3^-$ largely covaried in the ISOL-ICE ice core except between 1600 and 1800 when $MSA^-$ concentrations decreased while $NO_3^-$ concentrations increased. Concentrations of $MSA^-$ continued to increase over the extended two decades towards the present of the record. The mean $MSA^-$ and $NO_3^-$ values were 6.7 $\pm$ 2.3 ng g$^{-1}$ and 47 $\pm$ 13 ng g$^{-1}$ respectively.

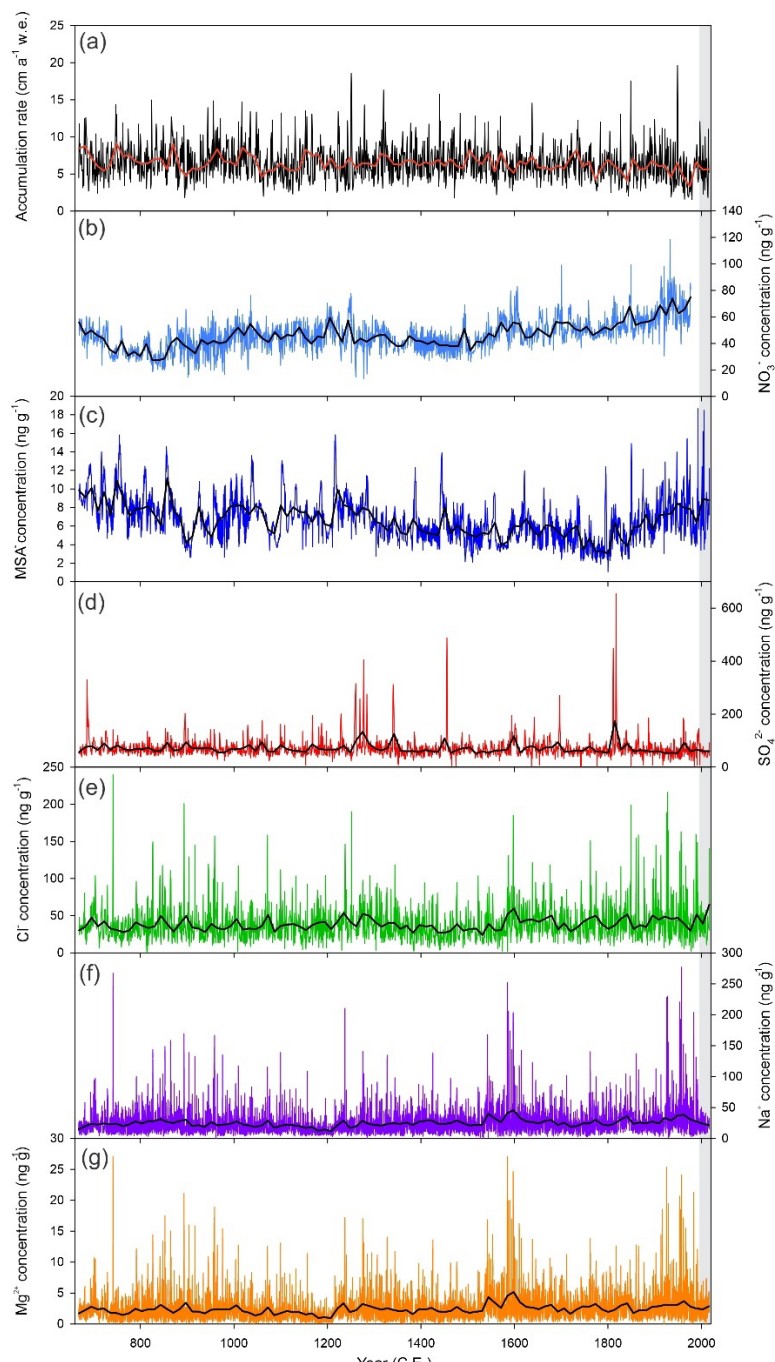

**Figure 3. Variability in (a) annual snow accumulation rate (b) nitrate (NO₃⁻) concentration, (c) methane sulphonate (MSA⁻)
270   concentration, (d) sulfate (SO₄²⁻) concentration, (e) chloride (Cl⁻) concentration, (f) sodium (Na⁺) concentration, and (g) magnesium
(Mg²⁺) concentration over the last millennium from the ISOL-ICE ice core. Black lines indicate smoothed data using a 0.01 loess
regression model (Cleveland and Devlin, 1988). Grey box: two-decade extension of the ice chemistry record from the Dronning Maud
Land site. Water equivalent: w.e. Common era: C.E.**

## 3.4. Meteorological conditions and spatial correlations

For the period 1957-2016, there are weak, but statistically significant at the 10 % level ($P$ <0.10), annual correlations found between SAM and $Na^+$ (R=-0.32) and accumulation (R=0.27). In addition, there are summer/early autumn (JFMA) statistically significant correlations for the SOI and $SO_4^{2-}$ (R=-0.26), i.e., La Niña (El Niño) is associated with lower (higher) $SO_4^{2-}$ concentrations, and Niño 3.4 and $MSA^-$ (R=0.26), and winter/spring Niño 3.4 and accumulation (R=0.24). The SAM relationships weaken and become insignificant after 1979. For the period 1979-2016, ENSO variability during summer/early autumn has moderately strong, statistically significant relationships with annual $Na^+$, $Mg^{2+}$, $MSA^-$, $Cl^-$, and $SO_4^{2-}$ variability, while the correlations are much weaker and insignificant during winter/spring (JJASON). Table A3 suggests that the underlying seasonality of the annual correlations between $SO_4^{2-}$ and ENSO is likely dominated by summer ENSO variability. The correlations are stronger for the shorter period of 1979-2016 compared to the longer 1957-2016 period.

Figure 4a shows the annual correlations for snow accumulation with 500 hPa geopotential height and meridional winds, 10 m meridional wind, mean sea level pressure and sea ice concentration for the period 1979-2016. Higher annual snow accumulation is associated with anomalous low pressure (anomalous cyclonic/clockwise circulation) over the Weddell Sea (significant in mean sea level pressure field) and northerly flow from the South Atlantic (significant at the 10 m level) bringing marine airmasses to the site (i.e., the negative correlation between meridional wind and accumulation implies positive accumulation anomalies are associated with northerly wind anomalies). The anomalous northerly flow corresponds with negative sea ice concentration anomalies in the outer edge of the sea ice field, with significant correlations in the northern Weddell Sea. While the northerly flow would directly contribute to a decrease in sea ice in the northern Weddell Sea through melt and/or convergence, it is also possible that the increase in open water could locally enhance atmospheric moisture content and contribute to higher snowfall.

Enhanced $MSA^-$ concentrations are associated with an anomalous high in the South Atlantic, an offshore low near 15°E, and an offshore high near 90°E (Figure 4b). In between the latter two pressure anomalies there is significant northerly flow onto the ice sheet between 30-45°E that circulates clockwise and flows offshore over the ISOL-ICE site due to the strong high-pressure anomaly in the South Atlantic. This onshore/northerly flow to the northeast of the site cuts across a local region of below average sea ice concentrations within the pack ice near Prydz Bay (~70°E), with mostly above average sea ice concentrations over the Weddell Sea where southerly flow prevails.

Seasonal correlations for $Mg^{2+}$ and $Na^+$ are similar as they are strongly positively correlated (R=0.65 for summer-autumn and R=0.97 for winter-spring). In winter, strong correlation and similar correlation maps in Figures 4 and A4) suggest that $Na^+$ and $Mg^{2+}$ are sea salt aerosol tracers from the open ocean or the sea ice and are advected with the same air masses (e.g. Wagenbach et al., 1998; Vega et al., 2018), whereas in summer the weaker correlation could be due to mineral dust contributions. During

the summer-autumn period high $Mg^{2+}$ and $Na^+$ is associated with a circumpolar zonal wave 3 pattern (Raphael, 2004) with three high-low pressure pairs encircling Antarctica. The zonal wave 3 pattern is most pronounced for $Mg^{2+}$. Within this pattern, there is anomalous northerly/onshore flow between 15-30°E bringing marine airmasses to the site. For Na, the onshore flow is tied to a local anticyclone just off the Enderby Land coast (near 45°E), and for $Mg^{2+}$ the onshore flow is tied to a local cyclone just off the DML coast (near 0°E). There is also significant northerly flow over the South Pacific associated with a cyclone north of the Ross Sea. The northerly flow from the South Pacific appears to cut across the West Antarctic Ice Sheet and flow offshore (turns to southerly wind anomalies/positive correlations) over the eastern Ronne-Filchner Ice Shelf (near 45°W), especially for $Mg^{2+}$. This transport pathway aligns with reduced sea ice concentrations in the Amundsen Sea which could contribute to enhanced $Mg^{2+}$ concentrations in summer-autumn.

During winter-spring, the large-scale circulation pattern and regional pressure anomalies associated with $Na^+$ and $Mg^{2+}$ are weaker and less coherent compared to summer-autumn. There appears to be two potential sources of onshore flow that bring marine air masses to the site in winter-spring. First, high $Mg^{2+}$ and $Na^+$ concentrations are associated with northerly flow over the Bellingshausen Sea/Antarctic Peninsula that cuts across Ellsworth Land and then flows off the continent along 0°E, which is weakly tied to an anticyclone north of the Weddell Sea and slightly lower-than-average pressure over the Amundsen Sea region. Second, similar to the annual $MSA^-$ correlations, there is a local cyclonic/clockwise circulation anomaly centred on the East Antarctic plateau near 10°E, which produces onshore northerly flow near 45°E that travels clockwise around the cyclone across the South Pole and reaches the ISOL-ICE site on southerly winds. The local southerly flow is enhanced by the weak positive pressure/anticyclonic circulation anomaly north of the Weddell Sea. There is a negative correlation between $Mg^{2+}$ and $Na^+$ concentrations and sea ice concentrations in the Weddell Sea around 0° yet this sea salt source region would not be tied to the ISOL-ICE ice core site as it is associated with offshore/southerly flow.

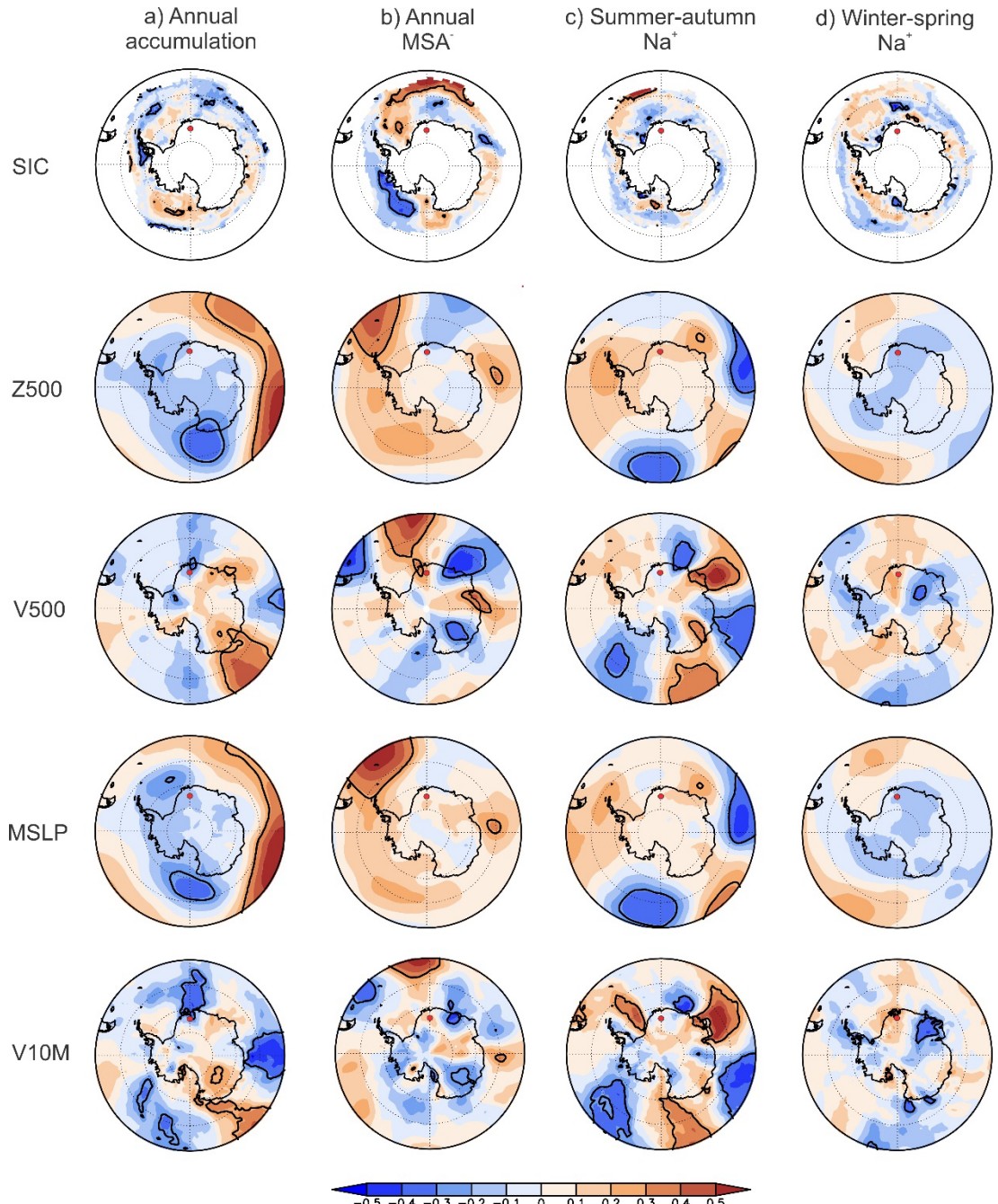

330 **Figure 4. Annual and seasonal correlations during the observational era (1979-2016) between the ISOL-ICE ice core record and sea ice concentration (SIC), 500-hPa geopotential height (Z500), 500-hPa meridional wind (V500), mean sea level pressure (MSLP), and 10 m meridional wind (V10M). Correlations are for (a) annual accumulation, (b) annual MSA⁻ concentration, (c) summer-autumn Na⁺ concentration, and (d) winter-spring Na⁺ concentration. The red dot is the location of the ISOL-ICE ice core site. Coloured shading shows Pearson's correlation coefficient values as indicated by the colour bar at the bottom. The bold black contours denote**
335 **correlations that are significant at the 10 % level based on a two tailed Students t-test.**

## 4. Discussion

### 4.1. Aerosol source regions

The winter sea salt peak in the ISOL-ICE ice core is dominated by a northerly atmospheric transport signal, that is northerly flow directly to the site or northerly flow in other Antarctic sectors with transport across the continent to the site, rather than an open water/sea ice source region signal. In the case of the ISOL-ICE site, direct transport of sea salt aerosol is most likely as wet deposition of aerosol is connected to lifting and adiabatic cooling of the air mass and precipitation events take four days to arrive at Kohnen Station from the Southern Atlantic Ocean consistent with the relatively short sea salt aerosol lifetime of a few days in the Southern Ocean (Landwehr et al., 2022; Reijmer et al., 2002). Comparison of ice core $SO_4^{2-}/Na^+$ ratios with the reference sea water ratio of 0.252 (Millero et al. (2008) and those from fractionated sea ice sources (< 0.252) may allow attribution of sea salt aerosol source regions assuming there are two sea salt sources from open ocean and sea ice regions (Abram et al., 2013). This approach is limited because only annual $SO_4^{2-}/Na^+$ ratios are retrieved from the ISOL-ICE core, yet the seasonally resolved snow pit reported by Winton et al. (2020) shows that the winter $Na^+$ maximum at the ISOL-ICE site is associated with lower $SO_4^{2-}/Na^+$ ratios (Figure A3). At the same time, nss-$SO_4^{2-}$ concentrations remain positive (Figures A3 and 6a). Thus, the ISOL-ICE winter $SO_4^{2-}/Na^+$ ratios > 0.252 and the positive nss-$SO_4^{2-}$ values both indicate enrichment compared to seawater. However, in winter $SO_4^{2-}/Na^+$ values decrease due to contributions from sea ice sources of sea salt aerosol such as blowing snow in the Antarctic (Figure A3) (Frey et al., 2020). At coastal Antarctic sites, such as Halley Station, the winter peak in $Na^+$ originates from sea salt aerosol production from blowing snow above sea ice in the Weddell Sea with minor contributions from sea-ice production and frost flowers in coastal leads in the Weddell Sea (Frey et al., 2020; Yang et al., 2019). The evidence for a sea ice source of sea salt at inland, high-elevation sites is harder to substantiate due to the higher nss-$SO_4^{2-}$ background levels from other sources relative to sea salt concentrations and thus negative values of nss-$SO_4^{2-}$ are often not observed (Abram et al., 2013). Analysis of year-round $SO_4^{2-}$ and $Na^+$ aerosol data from Kohnen Station also could not differentiate between open ocean and sea ice sources based on the fractionation of winter sea salt aerosol and its origin from a potential sea ice source, i.e., winter sea salt-$SO_4^{2-}$ makes a minor contribution to the annual total which is dominated by large summer nss-$SO_4^{2-}$ (Weller and Wagenbach, 2007). Therefore, the sea salt fractionation cannot be used as a chemical signature of sea ice source strength at the ISOL-ICE site because the nss-$SO_4^2$ contribution to the annual mean is so large. Whereas recent changes in the proximity of the ice front to ice core sites in coastal DML highlighted the important role of elevation of an ice core site on the sea salt record as observed from the calving of Trolltunga in 1950 which increased coastal sea salt concentrations and shifted nss-$SO_4^{2-}$ towards more negative values in a low elevation ice core (Vega et al., 2018). Yet the changing sea salt source region was not reflected in a higher elevation ice rise site in the region showing that, even proximal to the coast, elevation plays a role in the detection of a sea salt fractionation signal in the DML region. While the source of winter $Na^+$ at ISOL-ICE may be less certain, the correlation analysis can explain the northerly transport to the site.

In summer, sea salt deposition at the ISOL-ICE site is explained by a transport signal associated with northerly marine air masses from Enderby Land and DML and an open water aerosol source in the Amundsen Sea with atmospheric transport across West Antarctica. The low sea salt concentrations in summer occur with more enriched $SO_4^{2-}/Na^+$ values, and higher nss-$SO_4^{2-}$ and $MSA^-$ concentrations (R=0.15, $P$ <0.01) indicating biogenic $SO_4^{2-}$ aerosol from open water (Figure A3). The observed correlation is weak but statistically significant, meaning that the parameters are somehow related but the predictive skill to estimate one parameter from the other at a given point in time is very low. Taken together, at the seasonal scale, the correlation analysis and high-resolution ice chemistry provide evidence for sea salt variability driven by year-round northerly atmospheric flow, either directly to the site or from other regions of the continent that are transported to the site, with an additional contribution from an open ocean derived sea salt in summer rather than a sea ice source.

Over the last millennium, unfractionated sea salt from open ocean sources was relatively constant; $SO_4^{2-}/Na^+$ ratios remained enriched with a mean $SO_4^{2-}/Na^+$ of 3.7 ± 2.7 above the seawater value of 0.252 (Figure 5). A lack of $SO_4^{2-}$ depletion in the ISOL-ICE ice core from fractionated sea ice sources, likely dominated by blowing snow and contributions from frost flowers, is similarly not observed at other inland, high-elevation Antarctic ice core sites which is attributed to the low accumulation rate and low temporal resolution smoothing the winter signal, and high summer biogenic $SO_4^{2-}$ overriding the depleted $SO_4^{2-}$ signal (Rankin et al., 2000; Rankin et al., 2004). Furthermore, nss-$SO_4^{2-}$ from the open ocean does reach Kohnen Station in winter (Weller and Wagenbach, 2007). There are three notable periods in the ISOL-ICE sea salt chemistry record (Figures 4 and 6). The first is around 1200 where there are no extreme sea salt deposition events and $SO_4^{2-}/Na^+$ ratios become more enriched. The second is around 1600 where the frequency of sea salt deposition events increase. The third is from 1800 to present where there is an increase in the frequency of sea salt deposition events ($Na^+$, $Cl^-$ and $Mg^{2+}$ concentrations) at the same time as the increase in $MSA^-$ concentrations. Based on the sea salt-northerly wind relationship for the 1979-2016 period (Figure 4), the increase in extreme sea salt deposition years in the ISOL-ICE record likely reflects stronger sea salt transport rather than changing open water/sea ice conditions. We acknowledge that these relationships may change over time. For example, the SAM influence on Antarctic temperature and wind changed between 1957-1979 compared to 1979-present (Marshall et al., 2022). While the large-scale drivers of northerly wind may change, the sea salt-northerly wind relationship may hold true over the last millennium. Fundel et al. (2006) investigated $MSA^-$ variability in another core from the high-elevation Dronning Maud Land (DML05) in relation to atmospheric patterns. The authors found that over the reanalysis period (1969-1997), the Antarctic Dipole plays an important role during years of particularly high $MSA^-$ concentrations, while over the last 2000 years, periods of high $MSA^-$ concentrations are related to periods of higher sea-salt aerosol reflecting the influence of transport on both species despite the different seasonality. Atmospheric transport strength, rather than sea ice conditions, was also found by Abram et al. (2007) as the dominant factor determining the $MSA^-$ signal preserved in ice cores in the Weddell Sea sector. While longer-term trends in $MSA^-$ concentrations are not coupled with $SO_4^{2-}/Na^+$ ratios or sea salt variations throughout the ISOL-ICE record, transport clearly plays a dominant role in both records with the Antarctic Dipole likely aiding in the efficient atmospheric transport of both sea salt and $MSA^-$ to the site during periods for example, from 1200 to 1600 C.E.

The ISOL-ICE core reproduces the MSA⁻ and Cl⁻ records from existing DML cores (Fundel et al., 2006) and extends these records by two decades from 2000 to 2017. The striking feature of the extended MSA⁻ record is the continued increase in MSA⁻ concentration over the last two decades starting around 1800 (Figure 3). While MSA⁻ concentrations continue to increase over the last two decades, years of extreme sea salt deposition do not. Both changes in the seasonality and extent of regional sea ice, which impacts the source region and development of seasonal phytoplankton blooms in Antarctic waters (e.g. Abram et al., 2011), a change in DMS oxidation pathway from DMS to MSA and $SO_4^{2-}$, and a change in atmospheric circulation strength and pathway could contribute to the MSA⁻ increase over the last two centuries. During this same period of MSA⁻ increase, reconstructions of sea ice conditions from Antarctic ice cores show that sea ice extent declined in the Bellingshausen-Amundsen Sea region since 1850 C.E. (Dalaiden et al., 2021; Abram et al., 2010; Thomas et al., 2019) and the northernmost latitude of sea ice edge declined in the Weddell Sea sector (50°W–20°E) over the reconstructed period from 1900 to present (Yang et al., 2021). These recent changes call for an updated analysis of MSA⁻-sea ice relationships over the past two decades towards the present.

While previous studies found that MSA⁻ variability in the high-elevation DML region was associated with transport strength rather the sea ice conditions (Fundel et al., 2006; Abram et al., 2007), we investigated whether the additional two decades of MSA⁻ ice core data from ISOL-ICE captures local sea ice changes. Recently, Isaacs et al. (2021) and Clem et al. (2018) found that over the period 1979–2018, ENSO impacted summer and autumn sea ice concentration around coastal DML (10-70°E) where sea ice concentration increased during a cold ENSO phase and decreased during a warm ENSO phase. Here, sea surface temperature anomalies in the tropical Pacific Ocean influenced regional circulation over the South Atlantic and encouraged meridional airflow to DML affecting sea ice thermodynamically, by altering local heat transport and hence sea ice formation and melt. In the region where sea ice concentration is significantly correlated with ENSO, the sea ice concentration anomaly increased from 1979 to 2016 consistent with the pan Antarctic sea ice concentration trend, and also demonstrated that interannual variability was strongest in austral winter and spring where it was more subdued in summer and autumn when sea ice concentration was most strongly correlated with ENSO. Our correlation analysis (1979-2016) shows that ENSO variability during summer/early autumn contributes to the annual MSA⁻ variability and years of higher annual ice core MSA⁻ concentrations are associated with reduced sea ice concentrations west of Prydz Bay and this MSA⁻ aerosol is delivered to the ISOL-ICE ice core site by northerly distal winds. We note that the positive MSA-ENSO relationship is stronger over the shorter period of 1979-2016 compared to the 1957-2016 period which casts uncertainty over the long-term stability of the relationship. We did not find evidence for a locally derived MSA⁻ aerosol source in coastal DML (Figure 4b). Thus, our updated record shows that years of increased ice core MSA⁻ were likely driven by the transport direction with meridional winds that flow across a reduced sea ice region in the Prydz Bay region acting to synergistically reduce sea ice concentration while also increasing the delivery of MSA⁻ to the ISOL-ICE ice core site. Assuming the MSA⁻-wind direction relationship holds true for the last millennium, the increase in MSA⁻ since 1880 likely reflects stronger northerly transport over that period. Our

correlation analysis, combined with our finding that sea salts do not carry a fractionated sea ice signature over the last 1300 years, shows that sea salt aerosol and MSA$^-$ concentrations continue to be primarily related to atmospheric transport over the extended two-decade period.

While the variability of NO$_3^-$ concentration in the ISOL-ICE ice core will be discussed in a separate study, the significance of constraining aerosol source regions has implications for ice core NO$_3^-$. While NO$_3^-$ sources at remote high elevation Antarctic sites are not well constrained, the net primary sources in the Southern Ocean and Antarctica are peroxyacetyl nitrate (PAN) and methyl nitrate (CH$_3$NO$_3$) from open ocean and sea ice and stratospheric NO$_3^-$ (Burger et al., 2022; Burger et al., 2023; Mulvaney and Wolff, 1993; Jones et al., 2011). The similar temporal trends of NO$_3^-$ and MSA$^-$ compared to sea salts over the

last millennium suggests a potential link between MSA$^-$ and NO$_3^-$ through the contribution of oceanic CH$_3$NO$_3$ to ice core NO$_3^-$, i.e., the weak but statistically significant correlation between ice core NO$_3^-$ and MSA$^-$ (R=0.10, $P$ <0.01) is similar to nss-SO$_4^{2-}$ and MSA$^-$ (R=0.15, $P$ <0.01) where the latter relationship is traditionally used to assess biogenic sulfur contributions from the ocean.

Previous studies from the Antarctic coast and Weddell Sea region provide mixed evidence for a CH$_3$NO$_3$ source. For example, recent studies attribute some of the atmospheric NO$_3^-$ observed in the Weddell Sea boundary layer to oceanic alkyl nitrates including CH$_3$NO$_3$ (Burger et al., 2023) in addition to alkyl amide emissions from algae in the marginal sea ice zone around the Antarctic Peninsula (Brean et al., 2021; Dall'osto et al., 2019; Dall'osto et al., 2017). At Halley Station, Jones et al. (2011) observed atmospheric organic NO$_3^-$ (PAN and CH$_3$NO$_3$), dominate NO$_y$ in winter, and with a seasonality consistent with an

oceanic source. Weller et al. (2002) showed similar seasonality of NO$_y$ at Neumayer Station, yet they did not find any significant contribution from the marine boundary layer based on trajectory analyses and radioisotope measurements. However, at Halley Station, year-round NO$_3^-$ in surface snow correlated with inorganic NO$_y$ and not with CH$_3$NO$_3$ (Jones et al., 2011). The correlation may be different further from the source on the Antarctic plateau as suggested by the MSA$^-$ and NO$_3^-$ correlation in the ISOL-ICE ice core (full record: R=0.10, $P$ <0.01; record excluding 1600-1800: R=0.22, $P$ <0.01). As

there are no NO$_y$ observations in high-elevation Droning Maud Land, further field measurements and modelling studies are required to understand the relationship, to confirm an oceanic alkyl nitrate/CH$_3$NO$_3$ contribution to ice core NO$_3^-$ and whether these organic source changes are imprinted in the NO$_3^-$ concentration record, and to understand why the ice core MSA$^-$ and NO$_3^-$ relationship is intermittent i.e., between 1600-1800 when MSA$^-$ and NO$_3^-$ do not covary (R=0.24, $P$ <0.01). While sea salt fractionation at the ISOL-ICE ice core site cannot be used as a chemical signature of sea ice source strength, sea salt

concentrations show that oceanic aerosol sources have remained stable over the last millennium and the relationship between sea salts and reanalysis data products implies that there have been no large changes in atmospheric transport over the period. Therefore, trends in ice core NO$_3^-$ and its stable isotopic composition likely reflect changes in atmospheric processes such as photochemical recycling of NO$_3^-$ between the air-snow interface.

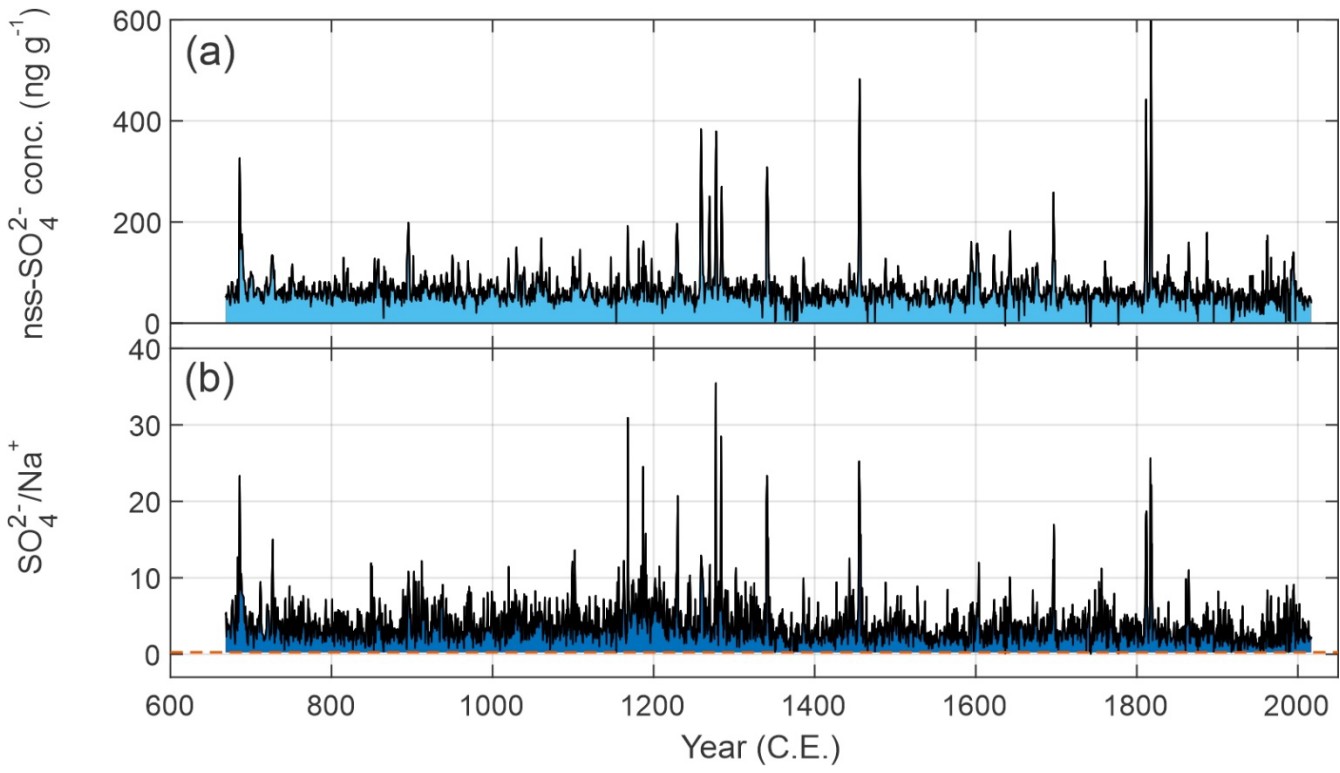


**Figure 5. Variability in ice core (a) non sea salt sulfate (nss-SO$_4^{2-}$) concentration, and (b) SO$_4^{2-}$/Na$^+$ ratio over the last millennium from the ISOL-ICE ice core. The orange dashed line indicates the SO$_4^{2-}$/Na$^+$ ratio of reference seawater. Above this line ice core SO$_4^{2-}$/Na$^+$ values represent open ocean sources in dark blue and below the line represents frost flower/sea ice sources of sea salt (Rankin et al., 2000). Positive nss-SO$_4^{2-}$ values in light blue indicate contributions from open ocean and sea ice margins, and negative**
**values indicate contributions from sea ice sources from blowing snow and frost flowers. The nss-SO$_4^{2-}$ contains all the volcanic peaks identified earlier for dating the ice core. Common era: C.E.**

## 4.2 Accumulation rate and implications for the nitrate isotope record

As local deposition influences the accumulation rate in the DML region as shown through multiple ice core accumulation records (e.g. Oerter et al., 2000; Sommer et al., 2000), a site-specific accumulation record is required for the interpretation of
the ISOL-ICE $\delta^{15}$N(NO$_3^-$) record. Snow accumulation at ISOL-ICE originates predominantly from the South Atlantic with onshore winds delivering marine air masses to the site (Figure 4) and the majority of precipitation occurring from frontal clouds (Reijmer and Oerlemans, 2002; Welker et al., 2014). This is consistent with air mass back trajectory studies by Reijmer et al. (2002) who show that air parcels and precipitation events at the Kohnen Station site originate in the Southern Atlantic Ocean around four days prior to their arrival on the DML plateau. Assuming the relationship between annual snow
accumulation and winds persisted over the last millennium, years of high snow accumulation in the ISOL-ICE record were driven by northerly atmospheric flow from the South Atlantic. The ISOL-ICE snow accumulation rate increased between 1979 and 1991 consistent with a composite accumulation record from ice cores in the region (Oerter et al., 2000). The increased accumulation rate is tied to local increases in northerly flow to the DML region associated with regional circulation changes

along/offshore the DML coast during the austral summer (December to February) and austral autumn (March to May) seasons. The broader circulation pattern trend in these two seasons resembles a zonal wave 3 pattern, especially in austral autumn. Thus, while we don't find a significant relationship between accumulation and ENSO or SAM, zonal wave 3, a well-known and prominent internal feature of the Southern Hemisphere atmospheric circulation (e.g. Raphael, 2004; Goyal et al., 2022), may be an important mechanism for producing localised northerly flow to the site that can influence accumulation variability. Also, these results reinforce earlier findings from Figure 4a and highlight the importance of sub-annual, seasonal circulation changes and variability in driving annual accumulation variability. Furthermore, other periods of decade long accumulation increases occur elsewhere in the record, e.g., between 1840 and 1849 which could also result from stronger synoptic circulation assuming the snow accumulation-geopotential height relationship did not change.

While the accumulation rate has been relatively stable over the last millennium and the record does not display centennial scale variability, the significant interannual variability in the record ranging between 2 and 20 cm a$^{-1}$ (w.e.) needs to be accounted for when interpreting the ice core $\delta^{15}N(NO_3^-)$ record which in turn is sensitive to changes in the accumulation rate and seasonality at the site (Winton et al., 2020). Air snow transfer modelling indicates that $\delta^{15}N(NO_3^-)$ preserved in the ISOL-ICE ice core will be less sensitive to changes in surface-UV than that retrieved at the lower snow accumulation site of Dome C. However, the higher snow accumulation rate and more accurate dating of the ISOL-ICE ice core allows for higher-resolution $\delta^{15}N(NO_3^-)$ based reconstructions of past surface-UV (TCO) (Winton et al., 2020). The 6 cm a$^{-1}$ average accumulation rate at the ISOL-ICE core site is high enough to preserve the seasonal cycle of $NO_3^-$ and $\delta^{15}N(NO_3^-)$ near the surface in large volume snow pit samples (Winton et al., 2020). The proportion of years in the accumulation record <6 cm a$^{-1}$ is 46 % and these low accumulation years improve the detection of multi-year trends in TCO, but at the expense of temporal resolution, i.e., when annual accumulation decreases below 6 cm a$^{-1}$, the seasonal $NO_3^-$ signal is not preserved, reducing noise in $NO_3^-$ and $\delta^{15}N(NO_3^-)$ from the impact of accumulation on $NO_3^-$ photolysis and concurrent isotope fractionation. Based on our age-depth model, the $NO_3^-$ concentrations in the ISOL-ICE core (Figure 3b) and the minimum $NO_3^-$ requirement for $\delta^{15}N(NO_3^-)$ using the preconcentration and denitrifier method (Morin et al., 2008), the highest resolution $\delta^{15}N(NO_3^-)$ record obtained from ISOL-ICE is annual rather than seasonal. This is higher resolution than what could be produced from the lower accumulation rate at Dome C (Winton et al., 2020), and is sufficient to achieve the goal of investigating the $\delta^{15}N(NO_3^-)$ variability over the last millennium. Future work will investigate the removal of the accumulation signature of the $\delta^{15}N(NO_3-)$ record to reconstruct past surface-UV radiation at the site using the inverse TRANSITS model which corrects for the effects of post-depositional processing on ice core $\delta^{15}N(NO_3^-)$ (Jiang et al., 2023).

Lastly, the ISOL-ICE accumulation rate falls within the range of 4 to 20 cm a$^{-1}$ for reconstructing surface mass balance from ice core $\delta^{15}N(NO_3-)$ proposed by Akers et al. (2022). The proportion of years in the accumulation record <4 cm a$^{-1}$ is 12 %. At these low accumulation rates, signal preservation due to drift, wind erosion, seasonal bias at the seasonal resolution is

compromised. Therefore, a surface mass balance reconstruction based on the Akers et al. (2022) rate transfer function would remove these very low accumulation years or if statistically not frequent smooth them by averaging a longer time scale.

## 5. Summary

In this study, we present the depth-age scale, accumulation rate and initial ice chemistry data from the 120 m deep ISOL-ICE ice core drilled at Kohnen Station in DML on the high-elevation Plateau in January 2017 to set the framework for the interpretation of the ISOL-ICE ice core record of stable nitrogen and oxygen isotopes in nitrate. The age scale was derived using well established dating techniques for ice cores in the region. The ISOL-ICE ice core spans the last 1349 years from 668 to 2017 C.E. $\pm$ 3 years extending the EDML core by 20 years towards the present. The mean snow accumulation rate of 6.5 $\pm$

2.4 cm a$^{-1}$ (w.e.) was stable over the last millennium, consistent with other ice cores in the region, with no notable changes over the last two decades. While the snow accumulation record does not display longer temporal trends over the last millennium, the significant interannual variability in the record needs to be accounted for when interpreting the nitrate stable isotope record which is sensitive to changes in the accumulation rate and seasonality at the site. Correlation of the snow accumulation record with climate variables over the observational era (1979-2016) showed that snow accumulation at ISOL-

ICE is predominately derived from the South Atlantic with onshore winds delivering marine air masses to the site.

    The ISOL-ICE core replicated other high-elevation DML ice chemistry records over the last millennium that were recovered decades earlier highlighting the reproducibility of climate and aerosol records in the region. Deposition of marine sea salts ($Na^+$, $Mg^{2+}$ and $Cl^-$) to the ISOL-ICE ice core site have remained stable over the last millennium. While the centennial scale

variability of $MSA^-$ and $NO_3^-$ concentrations largely covaried and increased over the last few centuries. The extended ISOL-ICE record of last two decades showed that $MSA^-$ continued to increase while there were less frequent deposition events of sea salts relative to the last millennium. Our correlation analysis, combined with our finding that sea salts do not carry a fractionated sea ice signature over the last 200 years, shows that sea salt aerosol and $MSA^-$ concentrations continue to be primarily related to atmospheric transport over the extended two-decade period. A potential link with $MSA^-$ and $NO_3^-$ ice core

variability warrants future work to investigate an oceanic alkyl nitrate/$CH_3NO_3$ contribution to ice core $NO_3^-$. As both sea salt deposition, atmospheric transport and snow accumulation rates remained stable over the last millennium, these factors are unlikely to drive the temporal variability on the observed 1300-year ISOL-ICE ice core record of $NO_3^-$ and its stable isotopic composition.

**Data availability** The datasets for the ISOL-ICE age-depth model, accumulation rate, DEP, and ice chemistry are available through the Polar Data Centre at https://doi.org/10.5285/9c972cfb-0ffa-4144-a943-da6eb82431d2 (Unreleased data) (Winton et al., 2019).

**Author contribution** Conceptualisation, M.M.F., J.S. and V.H.L.W.; methodology, M.M.F., R.M. and V.H.L.W.; validation, V.H.L.W.; formal analysis, V.H.L.W.; investigation, all authors.; resources, M.M.F. and R.M.; data curation, V.H.L.W.; writing - original draft preparation, V.H.L.W.; writing - review and editing, all authors; funding acquisition, M.M.F., J.S. and R.M. All authors have read and agreed to the published version of the manuscript.

**Competing interests** The authors declare that they have no conflict of interest.

**Acknowledgments** We would like to thank the British Antarctic Survey (BAS) and Alfred Wegener Institute (AWI) staff for their field and logistical support at Halley Station and Kohnen Station, respectively. Technical support for ice core measurements at BAS were provided by Rebecca Tuckwell, Lisa Hauge, Julius Rix, Catriona Sinclair, Emily Ludlow, and Shaun Miller. The ISOL-ICE ice core project was funded by a Natural Environment Research Council (NERC) Standard Grant (NE/N011813/1) to M.M.F. V.H.L.W. was supported by the NERC Standard Grant, and a Rutherford Foundation Postdoctoral Fellowship (RTF-VUW1801-PD) and Rutherford Discovery Fellowship (RDF-VUW2203) administered by the Royal Society Te Apārangi. K.R.C. acknowledges support from the Royal Society Te Apārangi Marsden Fund (MFP-VUW2010). J.S. thanks the French polar institute IPEV for its continuous support of field deployments and ice, snow and atmospheric sampling under the CAPOXI program (grant # 1177) and the European Union (grant 101054558 DOC-PAST). Views and opinions expressed are however those of the author(s) only and do not necessarily reflect those of the European Union or European Research Council (ERC). Neither the European Union nor the ERC can be held responsible for them. We thank Tessa Vance and an anonymous reviewer whose suggestions improved the paper.

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

**Appendices**

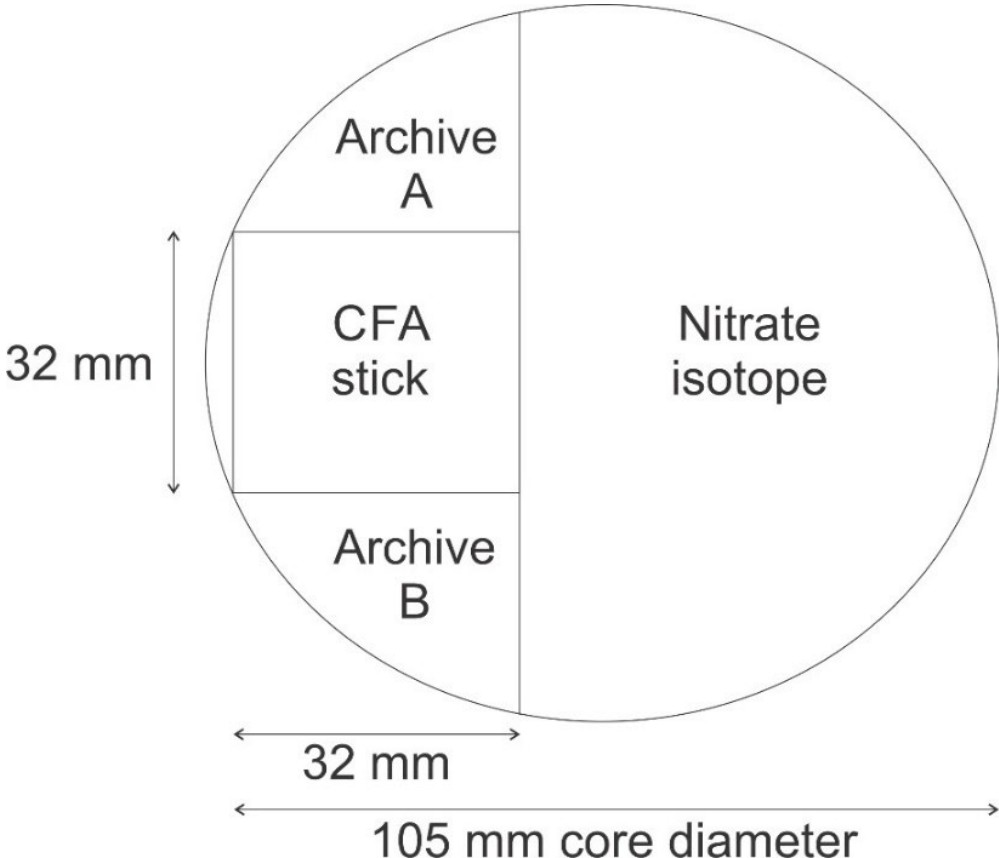

**Figure A1. Cross section of the ISOL-ICE ice core cut plan.**

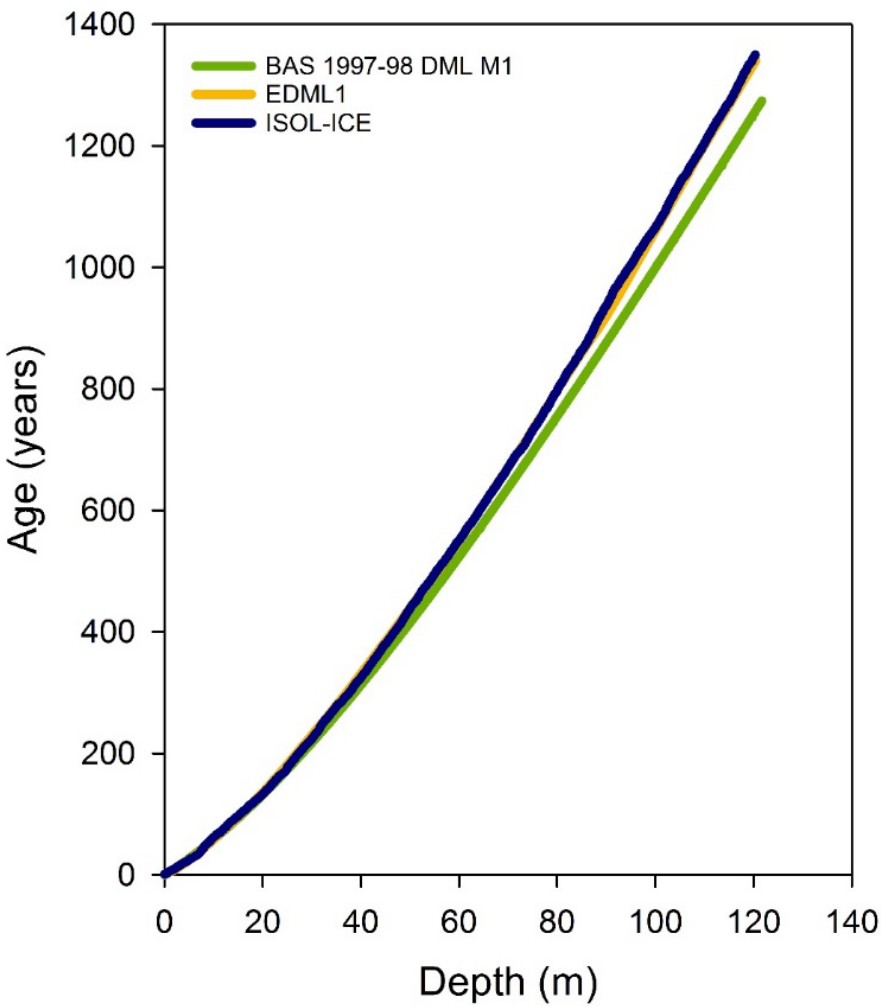

**Figure A2. Age depth model of the ISOL-ICE ice core and comparison to the EDML (EDML1 chronology (Ruth et al., 2007)) and M1 ice core (Hofstede et al., 2004). EDML and ISOL-ICE ice cores have the same age-depth scale as the drilling sites are located 1 km apart. Whereas, M1 was drilled in a slightly higher accumulation zone, and thus captures fewer years for the same depth.**

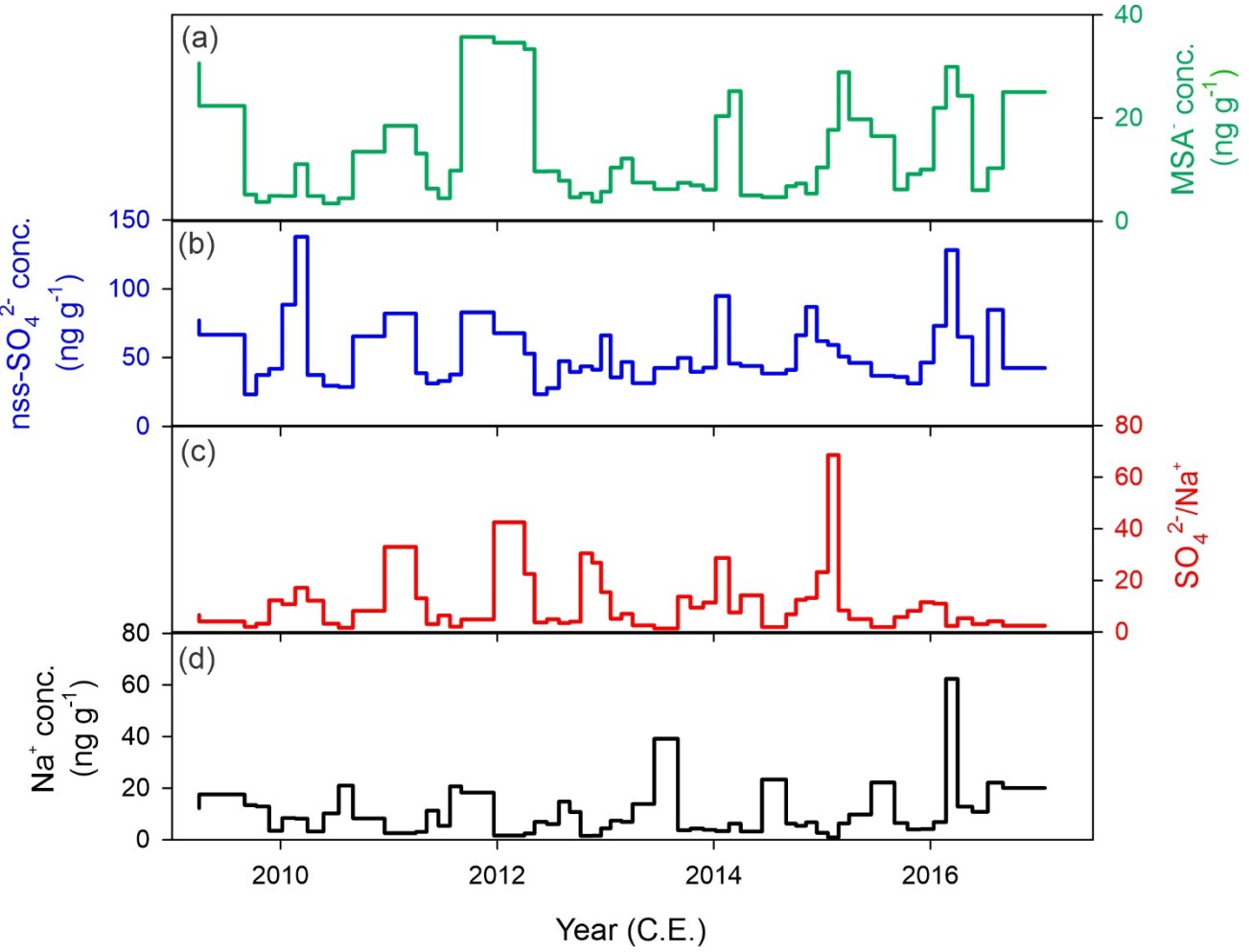


**Figure A3. Seasonal cycles in (a) methane sulphonate (MSA⁻), (b) non sea salt-sulfate (nss-SO₄²⁻), (c) SO₄²⁻/Na⁺ ratio, and (d) sodium (Na⁺) concentrations in ISOL-ICE snow pit A (Winton et al., 2020).**

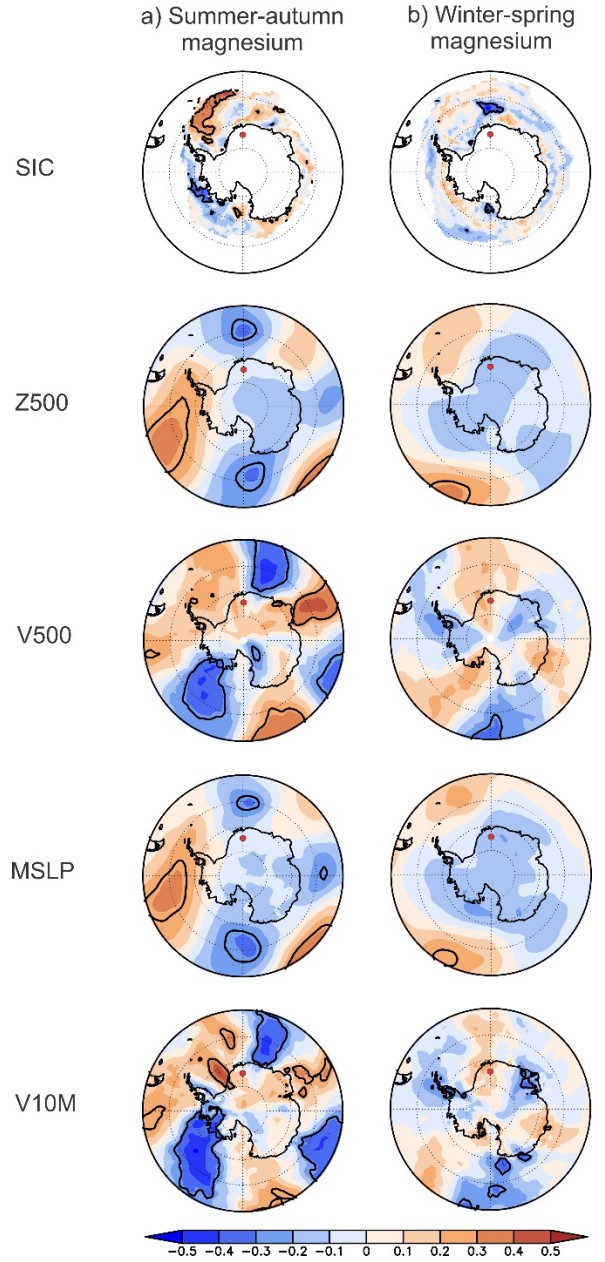

Figure A4. Seasonal correlations during the observational era (1979-2016) between the ISOL-ICE ice core record and sea ice concentration (SIC), 500-hPa geopotential height (Z500), 500-hPa meridional wind (V500), mean sea level pressure (MSLP), and 10 m meridional wind (V10M). Correlations are for (a) summer-autumn magnesium concentration, and (b) winter-spring magnesium concentration. The red dot is the location of the ISOL-ICE ice core site. Coloured shading shows Pearson's correlation coefficient values as indicated by the colour bar at the bottom. The bold black contours denote correlations that are significant at the 10 % level based on a two tailed Students t-test.

**Table A1. Accuracy and precision of soluble anion and trace element concentrations determined from repeated measurements of two CRM (European Reference Material ERM-CA408 simulated rainwater and European Reference Material, ERM-CA616 groundwater). Accuracy and precision are not reported for MSA⁻ as the CRM does not contain that analyte. The number of CRM measurements used for the accuracy and precision calculations are also listed (n).**

| Analyte | n | Accuracy (%) | Precision (%) |
|---|---|---|---|
| $Na^+$ | 20 | 94 | 4 |
| $Mg^{2+}$ | 20 | 113 | 4 |
| $NO_3^-$ | 20 | 107 | 10 |
| $SO_4^{2-}$ | 20 | 96 | 10 |

**Table A2.** Tie points for annual layer counting. The dates were taken from either the eruption date if known or the deposition date reported in the literature for ice cores in the DML region if unknown. ISOL-ICE deposition dates refer to the annually layer counted date at the identified volcanic horizon. No data: n.d.

| Tie points | Date (C.E.) | | | Depth (m) | |
|---|---|---|---|---|---|
| Drilling start | 2017 | | | 0 | |

| Eruption | Eruption date (AD) | Deposition date (AD) | ISOL-ICE deposition date (AD) | ISOL-ICE depth (m) | Reference |
|---|---|---|---|---|---|
| Pinatubo | 1991 | n.d. | 1991 | 4.91 | (Cole-Dai and Mosley-Thompson, 1999) |
| Krakatoa | 1883 | 1884 | 1885 | 19.75 | (Zielinski et al., 1994) |
| Tambora | 1815 | 1815 | 1816 | 27.38 | (Zielinski et al., 1994) |
| "1809" | 1809 | 1810 | 1809 | 28.02 | (Zielinski et al., 1994) |
| Komaga-Take | 1694 | 1695 | 1694 | 39.68 | (Zielinski et al., 1994) |
| Huaynaputina | 1600 | 1602 | 1601 | 48.13 | (Zielinski et al., 1994) |
| Kuwae | 1450 | 1457-58 | 1453 | 61.01 | (Plummer et al., 2012; Sigl et al., 2013; Cole-Dai et al., 2013) |
| "1285" | n.d. | 1285 | 1286 | 75.12 | (Langway et al., 1995) |
| "1277" | n.d. | 1277 | 1276 | 75.77 | (Langway et al., 1995) |
| "1259" | n.d. | 1259 | 1257 | 77.12 | (Langway et al., 1995) |
| "1228/30" | n.d. | 1229 | 1226 | 79.33 | (Langway et al., 1995) |
| "1166" | n.d. | 1168 | 1167 | 83.99 | (Zielinski et al., 1994; Langway et al., 1995) |
| "1108" | n.d. | 1108 | 1107 | 88.15 | (Traufetter et al., 2004) |
| "1050" | 1050 | 1040 | 1059 | 91.28 | (Traufetter et al., 2004) |
| "975" | n.d. | 975 | 968 | 98.45 | (Traufetter et al., 2004) |
| "961" | n.d. | 961 | 956 | 99.52 | (Traufetter et al., 2004) |
| "719" | n.d. | 719 | 720 | 116.40 | (Traufetter et al., 2004) |
| "685" | n.d. | 685 | 684 | 118.90 | (Traufetter et al., 2004) |


**Table A3. Correlation coefficients of annual ice core chemistry and snow accumulation rate with the SAM, Niño 3.4, and SOI indices over the period 1957-2016 and 1979-2016. Correlations significant at p < 0.10 are boldface. The significance is calculated based on a two-tailed Students t-test. The monthly boundaries for calculating the indices are January to December for SAM and January to April and June to November for Niño 3.4 and SOI.**

| | Seasonal boundary | Na$^+$ | Mg$^{2+}$ | MSA$^-$ | Cl$^-$ | SO$_4^{2-}$ | Accumulation |
|---|---|---|---|---|---|---|---|
| 1957-2016 | | | | | | | |
| SAM | January-December | **-0.32** | -0.2 | -0.08 | -0.17 | -0.02 | **0.27** |
| Niño 3.4 | January-April | 0.11 | 0.11 | **0.26** | 0.19 | 0.14 | -0.01 |
| Niño 3.4 | June-November | -0.11 | -0.02 | -0.09 | 0.01 | 0.05 | **0.24** |
| SOI | January-April | -0.17 | -0.19 | -0.20 | -0.20 | **-0.26** | -0.01 |
| SOI | June-November | 0.09 | 0.01 | 0.10 | 0.00 | -0.16 | -0.08 |
| 1979-2016 | | | | | | | |
| SAM | January-December | -0.14 | 0.08 | 0.03 | -0.16 | -0.08 | 0.21 |
| Niño 3.4 | January-April | **0.36** | **0.38** | **0.37** | **0.34** | **0.33** | -0.06 |
| Niño 3.4 | June-November | -0.22 | -0.05 | 0.01 | -0.13 | 0.20 | 0.23 |
| SOI | January-April | **-0.42** | **-0.43** | **-0.33** | **-0.29** | **-0.43** | 0.09 |
| SOI | June-November | 0.14 | 0.01 | -0.04 | 0.06 | **-0.36** | -0.06 |
