# Peer review of "Drivers of late Holocene ice core chemistry in Dronning Maud Land: The context for the ISOL-ICE project"

_Climate of the Past, 2023_

## Referee Comment (RC2)

Review for Winton et al., Climate of the Past: *Drivers of late Holocene ice core chemistry in Dronning Maud Land: The context for the ISOL-ICE project*

By V. Holly L. Winton, Robert Mulvaney, Joel Savarino, Kyle R. Clem, Markus M. Frey

This manuscript outlines the development of the age scale and associated chemistry records for the ISOL-ICE project. The manuscript looks at standard kinds of atmospheric and index correlations to understand the climatology driving the site specific characteristics. The manuscript is clear and readable, and well referenced. I have a few concerns (outlined below) which will likely require a bit of extra analysis, however I recommend the manuscript for publication once these are addressed. I have selected major revisions, but it is somewhere between minor and major revisions as what I suggest to do is not too onerous and I doubt will change the conclusions hugely, but will make the methods more defensible and easier to interpret.

**Major comments:**

I think the abstract needs a final sentence or two explaining where this work fits toward the eventual quantification of natural variability in ozone. At the moment the opening of the abstract doesn't follow to the results and conslcusions section of the abstract very well.

What proportion of years in the accumulation record are less than 6 cm (the minimum to preserve the nitrate seasonal cycle cited at line 50)? What effect will low accumulation years have on the ability to interpret the nitrate record for ozone variability? Also, it might be good to explain a bit more the need for at least 6 cm per annum to preserve the nitrate seasonal cycle, and that Akers et al., 2022 cite 4-20 cm. Finally, its not clear to me how a sampling resolution of 0.3 to 0.5 years provides a robust seasonal cycle – that is only 2-3 samples per year. How will this affect how you interpret the nitrate record? Also I suggest you change sentence at line 50 to 'higher rates' or similar rather than 'increases'. Reading further – perhaps I have misunderstood the sampling resolution for different analytes, so maybe this could be clarified with a table (see comment below).

I'd recommend a schematic in the introduction that outlines how Winton et al. 2020 and this work sit in the larger frame of eventually being able to robustly interpret the stable isotopic composition of nitrogen in snow and ice as a TC0/ozone layer proxy. At the moment, its not totally clear from the abstract and the introduction what the prior work has done, what this manuscript does, and what the future work will be, and it would be good to have that clarified at the start of the manuscript.

Can you expand a bit on volcanic dates – specifically the delayed 'arrival date' of Huaynaputina (2 years) and Kuwae (8 years)? This seems a bit odd with an error of 3 years.

Lines 75-77 – but how do you do this faithfully with only 2-3 measurements per year? You need some caveats here that your sample resolution limits your ability to interpret sub-annually here. Perhaps detail what sample resolution you get in the surface cores compared to at depth. Now that I have got to the results, I see at lines 218-219 I might have missed detail on what the sample resolution is for different analyses? If so, perhaps explain with a table (as I suggest above) the sample resolution for each type of analyste both at the surface and at depth so the reader can clearly see where you get seasonal / sub-annual resolution and where you don't.

It isn't clear to me why you used a calender year (Jan-Dec?) mean for the ENSO indices – is that correct? If so, this would split any ENSO event in half and partition it to separate years. This doesn't make sense to me from a climatological perspective. II'd strongly suggest using a more appropriate seasonal split of either April-March or May to April (if you want to use a full 12 month mean). You could also use June to Feb. Alternatively, you could use June-November, as the ENSO oceanic and atmospheric anomalies that are relevant to high southern latitudes commence around May / June, and intensify through to early summer, by which stage any event is well established. E.g. see Crockart et al., 2021 for an example of different seasonal splits. Lines 210-214 – this may well need a re-write once you have more appropriate seasonal boundaries for the Niño 3.4 and SOI

indices. For table A3 please add in caption what monthly boundaries you end up using for calculating the indices.

Figure 4 and Discussion. A dot on each map for the ice core site would be helpful to orientate the reader. As you state, there is not a huge difference between the magnesium and sodium maps (because the records are very highly correlated, especially winter spring). Do you need to present both? At the moment its hard to interpret any differences, especially for winter spring as they may be artefacts of analysis more than anything. Line 322 – what about orographic factors and relatively short sea salt aerosol lifetimes of a few days? Can you prove transport across the continent, versus say short-term episodic inputs of high sea salt loads that can't be teased out in your 6 month means? I'm not suggesting you have to prove this, but I think the across continent statement is a big one and lacks evidence in this context, especially given the statement about air mass / accumulation source at lines 459-461. Would that mean that any cross-continental sea salt trabsport was dry deposited?

**Minor comments:**

In the abstract, I'd recommend separating into two sentences the sentence about the development of the age model, and then the snow accumulation and ice chemistry records and correlations. These are two separate (fairly major) steps and deserve a sentence each.

Line 27 – over the last two decades.
Line 66 – independently derived – do you mean via layer counting and volcanic horizons?
Line 279 – figure 4b? also, what is an 'offshore low' and an 'offshore high'?
Line 369 – sea salt
Line 390 – what do you mean by a change in the DMS oxidation pathway?

---

## Author Comment (AC1)

Thank you for the thoughtful and helpful reviews to strengthen the paper.

**Reviewer comment 1**

In this paper by Winton and co-authors the results achieved from a 120 m deep ice-core are presented. The ice-core was drilled in the DML region (close to Kohnen Station) in 2017 in the framework of the ISOL-ICE project. The main results shown and described in the paper can be summarised in: production of a chronology for the last 1349 years, the history of the accumulation rate at the site along this period and the chemical profiles of selected markers (mainly nitrate, MSA and sea spray species). The chronology was built via annual layers counting with the aid of selected accurate age markers to be used as tie points (volcanic eruptions). The authors clearly write that this work is just the first part of the full job to be done in the framework of the project and I agree with them that the reconstruction of the accumulation rate is very important to better understand the nitrogen isotopes variations in nitrate. We all agree on the importance of the study of the TCO variations in the past but, in this paper, this topic is not studied at all and some sections of the Abstract and of the Introduction should be "cleared" at least in part since they are a bit misleading. I think that especially the Abstract should not give all those details about TCO, UV radiation and nitrogen isotopes. The discussion about the importance of this topic is properly described in section 4.2.

In the abstract and introduction, we have reduced the background information about the ozone layer and UV radiation. We have left the information in the abstract and introduction about the accumulation rate-nitrate isotope relationship as quantifying the accumulation variability and the implications for nitrate isotope interpretation at the site is one of the motives of the study. We have addressed the comments of reviewer 2 about explaining where the work presented here fits within the development of a UV (ozone) proxy in the introduction and a final sentence explaining where the results of this manuscript fit towards the eventual quantification of natural ozone variability in the abstract.

Regarding the chronology, the authors declare an estimated uncertainty at the bottom of the core of 3 years. I have two questions for this point:

1. How was the uncertainty estimated? Is it a sum of the "uncertain" annual layers?

   Yes, it was the sum of uncertain annual layers counting from the previous volcanic age marker.

2. Since you have several well-known volcanic age markers, I would expect the maximum uncertainty in the middle of the largest interval enclosed between volcanic peaks. Which is the maximum uncertainty along the core? Does it coincide with the bottom uncertainty?

   Thank you for this important comment. We agree that reporting the uncertainty over the largest interval between dated volcanic peaks is more appropriate. A maximum uncertainty of 3 years was estimated based on the largest difference in the number of years between a documented eruption date and ISOL-ICE deposition date. To determine this, we compared dates of volcanic deposition horizons with documented eruption dates (Pinatubo, Tambora, Krakatoa, Huaynaputina, Komaga-Take, Kuwae, Samalas). These

eruptions are the most established and well-known volcanic markers observed in Antarctic ice cores over this period.

L250-251 *"A maximum age uncertainty in our layer counting of ± 3 years is estimated between 48-61 m corresponding to about 150 yr (1453-1601 AD), which is one of the largest intervals enclosed between documented volcanic peaks in Antarctic ice cores (Table A2) ."*

All along the manuscript the authors use Na and Mg but from time to time they use the ionic form. I know that the authors measured the "total" content of the two species but, as stated at line 139, the total is assumed to represent mostly the ionic specie. The same thing holds true for chloride which is sometimes Cl- and other times Cl. I think that the ionic forms would be to be preferred all along the text.

We have updated the manuscript with the ionic form throughout.

Figure 3 (a) shows an interesting "dip" a few years before 2000 but this feature is not discussed in the text. Do the authors have a possible explanation for this section? Such significant variations in the accumulation rate will be basic in correctly interpret the N-nitrate isotopic changes and thus past surface-UV and TCO implications.

Thank you for pointing out this interesting feature in the record. The ISOL-ICE snow accumulation rate decreases between 1950-1980 and increases between 1979 and 1991. Figure A below shows the running 13-year trends for the accumulation rate and shows that the 1979-1991 period had the strongest accumulation increase on this decadal time scale. The increase in accumulation over the period is consistent with a composite accumulation record from ice cores in the region (Oerter et al., 2000). Reanalysis data prior to this is less reliable over Antarctica and the Southern Ocean as this is a data spare region and the satellite sounder data were assimilated from 1979 onward. Thus, we examined the 1979-1991 period further.

[Figure]

*Figure A: Running 13-yr trends of the ISOL-ICE accumulation rate, showing the 1979-1991 period has the strongest accumulation increase on this decadal time scale.*

Figure B shows the 500 hPa geopotential height and wind annual and seasonal trends for the 1979-1991 period. In terms of annual trends, there was stronger cyclonic circulation. There was also increased northerly flow to the site in certain seasons. Figure B shows more northerly flow coming in from the Weddell Sea to the west of the ISOL-ICE site in autumn (MAM) tied to increased pressure/anticyclonic circulation along the coast of DML near 0° longitude. This can explain the higher accumulation in MAM and thus dominating the increase in the annual accumulation rate. In addition, there was more northerly flow to the ISOL-ICE site from the northeast in summer (DJF) tied to a decrease in pressure/cyclonic circulation offshore of DML between 0-15°E. Both circulation trends in DJF and MAM reflect a trend toward a more prominent zonal wave 3 pattern (three high-low pressure pairs surrounding Antarctica), especially in MAM.

The finding over the 1979-1991 period reinforces the correlations observed for annual accumulation in Figure 4a and our conclusions that higher annual snow accumulation is associated with anomalous low pressure over the Weddell Sea and northly flow from the South Atlantic bringing marine airmasses to the ice core site. While we did not find any significant changes with temperature, ENSO or SAM over this period, other periods of decade long accumulation increases occur elsewhere in the record. For example, between 1840 and 1849 which could be tied to synoptic circulation assuming the snow accumulation-geopotential heigh relationship did not change.

Text added to L505-516 *"The ISOL-ICE snow accumulation rate increased between 1979 and 1991 consistent with a composite accumulation record from ice cores in the region (Oerter et al., 2000). The increased accumulation rate is tied to local increases in northerly flow to the DML region associated with regional circulation changes along/offshore the DML coast during the austral summer (December to February) and austral autumn (March to May) seasons. The broader circulation pattern trend in these two seasons resembles a zonal wave 3 pattern, especially in austral autumn. Thus, while we don't find a significant*

*relationship between accumulation and ENSO/SAM, zonal wave 3, a well-known and prominent internal feature of the Southern Hemisphere atmospheric circulation (e.g. Goyal et al., 2022; Raphael, 2004), may be an important mechanism for producing localised northerly flow to the site that can influence accumulation variability. Also, these results reinforce earlier findings from Figure 4a and highlight the likely importance of sub-annual, seasonal circulation changes and variability in driving annual accumulation variability. Furthermore, other periods of decade long accumulation increases occur elsewhere in the record, e.g., between 1840 and 1849 which could also result from stronger synoptic circulation assuming the snow accumulation-geopotential height relationship did not change."*

Text added L497-499 *"As local deposition influences the accumulation rate in the DML region as shown through multiple ice core accumulation records (e.g., Oerter et al., 2000; Sommer et al., 2000), a site-specific accumulation record is required for the interpretation of the ISOL-ICE $\delta^{15}N(NO_3^-)$ record."*

**1979 -1991 500  hPa Geopotential Height & Wind Trends**

*Figure B: The 500 hPa geopotential height and wind trends for annual and seasonal periods between 1979 and 1991. Bold contours outline 10 % significance level trends, and wind vectors are plotted only if at least one wind component trend is significance at the 10 % level.*

Minor comments:

Line 19: when the authors speak about an extension of the previous records by two decades, I guessed that was an extension back in time but I understood later that it's an extension towards more recent years. Elsewhere in the manuscript this was correctly said. I invite the authors to correct this expression in the abstract in order to clarify this point.

Text added L20 "…towards the present."

Line 132: insert a space between Cl- and "was".

Done.

Line 139:  change to "are assumed"

Done.

Line 263: change "with SAM …" to "between SAM…"

Done.

Line 488: change the full stop before "While" in a comma.

We have left the sentences as two separate sentences combining them results in a very long sentence.

Figure A3: bottom panels: y-axes add + to Na in both panels

Done.

Table A2: I would prefer "Deposition date" instead of "Arrival date" in the title of the third column.

Done.

Table A3: the SAM/accumulation R should be bold
Done.

**References**

Goyal, R., Jucker, M., Gupta, A. S., and England, M. H.: A new zonal wave-3 index for the Southern Hemisphere, Journal of Climate, 35, 5137-5149, 2022.
Oerter, H., Wilhelms, F., Jung-Rothenhäusler, F., Göktas, F., Miller, H., Graf, W., and Sommer, S.: Accumulation rates in Dronning Maud Land, Antarctica, as revealed by dielectric-profiling measurements of shallow firn cores, Annals of Glaciology, 30, 27-34, 2000.
Raphael, M.: A zonal wave 3 index for the Southern Hemisphere, Geophysical Research Letters, 31, 2004.

---

## Author Comment (AC2)

We would like to thank the referee for the time taken and the thoughtful suggestions on our manuscript. They have been an asset to the study and have led to an improved manuscript.

Reviewer comment 2

This manuscript outlines the development of the age scale and associated chemistry records for the ISOL-ICE project. The manuscript looks at standard kinds of atmospheric and index correlation to understand the climatology driving the site specific characteristics. The manuscript is clear and readable, and well referenced. I have a few concerns (outlined below) which will likely require a bit of extra analysis, however I recommend the manuscript for publication once these are addressed. I have selected major revisions, but it is somewhere between minor and major revisions as what I suggest to do is not too onerous and I doubt will change the conclusions hugely, but will make the methods more defensible and easier to interpret.

Major comments:

I think the abstract needs a final sentence or two explaining where this work fits toward the eventual quantification of natural variability in ozone. At the moment the opening of the abstract doesn't follow to the results and conclusions section of the abstract very well.

Text added in L32-33 *"...and thereby leaving the UV imprint in the $\delta^{15}N(NO_3^-)$ ice core record to quantify natural ozone variability."*

What proportion of years in the accumulation record are less than 6 cm (the minimum to preserve the nitrate seasonal cycle cited at line 50)? What effect will low accumulation years have on the ability to interpret the nitrate record for ozone variability? Also, it might be good to explain a bit more the need for at least 6 cm per annum to preserve the nitrate seasonal cycle, and that Akers et al., 2022 cite 4-20 cm. Finally, its not clear to me how a sampling resolution of 0.3 to 0.5 years provides a robust seasonal cycle – that is only 2-3 samples per year. How will this affect how you interpret the nitrate record? Also I suggest you change sentence at line 50 to 'higher rates' or similar rather than 'increases'. Reading further – perhaps I have misunderstood the sampling resolution for different analytes, so maybe this could be clarified with a table (see comment below).

The proportion of years in the accumulation record $<6$ cm a$^{-1}$ is 46 %. When annual accumulation decreases below 6 cm a$^{-1}$ the seasonal $NO_3^-$ signal is not preserved, reducing noise in $NO_3^-$ and $\delta^{15}N(NO_3^-)$ from the impact of accumulation on nitrate photolysis and concurrent isotope fractionation and therefore rather improves detection of multi-year trends in TCO, but at the expense of temporal resolution.

The 4-20 cm a$^{-1}$ accumulation rate range refers to the accumulation rate of ice core sites that are suitable for reconstructing surface mass balance from $\delta^{15}N(NO_3^-)$ due to the strong inverse relationship of $\delta^{15}N(NO_3^-)$ and accumulation across the continent as proposed by Akers et al. (2022). This is different to the 6 cm a$^{-1}$ average accumulation rate at the ISOL-ICE core site which is high enough to preserve a seasonal cycle of $NO_3^-$ and $\delta^{15}N(NO_3^-)$ (6 cm a$^{-1}$ water equivalent is around 30 cm of snow, largely enough to correctly sample annual layers near the surface). It was the large volume snow pit samples in Winton et al. (2020) that allowed us to seasonally resolve the nitrate isotope record. While we produced the highest possible resolution nitrate isotope record from the ISOL-ICE core (not published), the

resolution is annual rather than seasonal due to the lower sample volume. This is still significantly higher resolution than what could be produced from Dome C, and sufficient to achieve the goal of investigating the nitrate isotope variability over the last millennium. Therefore, the large proportion of years $<6$ cm $a^{-1}$ is irrelevant in the case of the ISOL-ICE core.

However, the proportion of years in the accumulation record $<4$ cm $a^{-1}$ is 12 %. At these low accumulation rates, signal preservation (drift, wind erosion, accumulation patch, seasonal bias) at the seasonal resolution can be compromised. Therefore, an accumulation rate reconstruction based on the Akers et al. (2022) rate transfer function, should remove these very low accumulation years or if statistically not frequent smooth them by averaging longer time scale.

Apologies for the confusion over the sample resolution. We have edited the text with the average number of data points per year at the top and bottom of the core. Due to the high sample resolution of the chemical markers used for annual layer counting (e.g., sodium and magnesium), we can confidently interpret the record on a sub-annual resolution. The lower sample resolution of 0.3-0.5 years refers to the soluble ion data which was not used to date the core.

L162-164 *"The sampling resolution of the ICP-MS, liquid conductivity and insoluble particle measurements was <1 mm resulting in an average of 190 ± 120 measurements per year in the top 5 m of the core and an average of 140 ± 60 measurements per year in the bottom 5 m of the core."*

L144-145 *"The sampling resolution of the FIC measurements was 5 cm at the top of the core and 4 cm at the bottom resulting in an average of 2 ± 1 measurements per year."*

L53 "increases" replaced with "*higher rates.*"

I'd recommend a schematic in the introduction that outlines how Winton et al. 2020 and this work sit in the larger frame of eventually being able to robustly interpret the stable isotopic composition of nitrogen in snow and ice as a TC0/ozone layer proxy. At the moment, its not totally clear from the abstract and the introduction what the prior work has done, what this manuscript does, and what the future work will be, and it would be good to have that clarified at the start of the manuscript.

Thank you for this suggestion to clearly outline the development process of the UV/TCO ice core proxy.

The first step is an air-snow transfer study where high resolution (daily and seasonal) observations of nitrate concentration and isotopes were made at the ice core site. These observations were used to quantify the effect of the accumulation rate, light attenuation and TCO on the archived snow $\delta^{15}N(NO_3^-)$ signature using the TRANSITS model. The results, published in Winton et al. (2020), show that the accumulation rate along with e-folding depth are the dominant factors controlling the archived $\delta^{15}N(NO_3^-)$ signature, followed by TCO and accumulation timing. The second step is the development of the age scale, accumulation rate and associated chemistry records for the ice core. These results, presented in this manuscript, provide foundational data to understand nitrate source regions and the natural variability of changes in the site specific accumulation rate. The final step is to use the ice core nitrate

concentration and accumulation rate record reported in this study, along with the measured ice core $\delta^{15}N(NO_3^-)$, to model the site specific surface conditions using the inverted TRANSITS model (Jiang et al., 2023). The UV (ozone) proxy is extracted from the $\delta^{15}N(NO_3^-)$ record by constraining the accumulation rate effect on the $\delta^{15}N(NO_3^-)$ thereby leaving the UV effect. Application of the UV (TCO) transfer function to the ice core $\delta15N(NO3-)$ which will be presented in a future study.

In L84-96, we believe we have now clearly incorporated these steps into the introduction as thus a schematic is not necessary.

Can you expand a bit on volcanic dates – specifically the delayed 'arrival date' of Huaynaputina (2 years) and Kuwae (8 years)? This seems a bit odd with an error of 3 years.

Thank you for pointing this out. We used published volcanic dates previously used to reconstruct well established age-depth models for ice cores in the DML region. We apologise for the confusion which likely arises around our explanation of the chosen dates which were taken from either the eruption date if known or ice core arrival (=deposition) dates in the literature if unknown. The deposition date in Table A2 refers to the published deposition date from ice cores in the literature rather than the ISOL-ICE deposition dates. For example, common observation of the layer attributed to Kuwae in other ice cores is around 1457-1458 (deposition date; Plummer et al. (2012); (Cole-Dai et al., 2013; Sigl et al., 2013), and the commonly accepted eruption date is 1450 (eruption date). The ISOL-ICE deposition date for Kuwae is 1453.

We have now added the deposition date for the ISOL-ICE core to the Table A2 which was accidentally omitted in the submitted version of the manuscript. We apologise for the confusion. We have edited the table caption and added text around the volcanic dates to make this clearer. Please see response to reviewer 1 concerning at dating uncertainty of 3 years.

L179-182 *"The volcanic dates were taken from either the eruption date if known or published deposition dates used to construct age-depth models for ice cores in the DML region if unknown (e.g. Cole-Dai and Mosley-Thompson, 1999; Zielinski et al., 1994; Langway et al., 1995; Traufetter et al., 2004)."*

Lines 75-77 – but how do you do this faithfully with only 2-3 measurements per year? You need some caveats here that your sample resolution limits your ability to interpret sub-annually here. Perhaps detail what sample resolution you get in the surface cores compared to at depth. Now that I have got to the results, I see at lines 218-219 I might have missed detail on what the sample resolution is for different analyses? If so, perhaps explain with a table (as I suggest above) the sample resolution for each type of analyste both at the surface and at depth so the reader can clearly see where you get seasonal / sub-annual resolution and where you don't.

Please see response above regarding the sample resolution.

It isn't clear to me why you used a calendar year (Jan-Dec?) mean for the ENSO indices – is that correct? If so, this would split any ENSO event in half and partition it to separate years. This doesn't make sense to me from a climatological perspective. I'd strongly suggest using a more appropriate seasonal split of either April-March or May to April (if you want to use a full 12 month mean). You could also use June to Feb. Alternatively, you could use June-

November, as the ENSO oceanic and atmospheric anomalies that are relevant to high southern latitudes commence around May / June, and intensify through to early summer, by which stage any event is well established. E.g. see Crockart et al., 2021 for an example of different seasonal splits. Lines 210-214 – this may well need a re-write once you have more appropriate seasonal boundaries for the Niño 3.4 and SOI indices. For table A3 please add in caption what monthly boundaries you end up using for calculating the indices.

Thanks for the great suggestion to examine ENSO variability based on its seasonal cycle. We have accordingly correlated the annual data with July to April (JFMA) and June to November (JJASON; Crockart et al., 2021) seasonal-mean ENSO variability noting that May is generally the transition month between ENSO cycles. We still find that annual accumulation shows no significant relationship with ENSO variability. However, we find that ENSO variability during JFMA (summer/early autumn) has moderately strong, statistically significant relationships with annual $Na^+$, $Mg^{2+}$, $MSA^-$, $Cl^-$, and $SO_4^{2-}$ variability, while the correlations are much weaker and insignificant during JJASON (winter/spring). This helps shed light on the underlying seasonality of the annual correlations seen between $SO_4^{2-}$ and ENSO shown in Table A3, in that these are likely dominated by summer ENSO variability. However, it is interesting to note that the correlations are stronger for the shorter period of 1979-2016 compared to the longer 1951-2016 period, thus may cast some uncertainty over the long-term stability of this relationship. Nevertheless, we can conclude from this analysis over the modern observation era that ENSO variability during summer does appear to be an important driver of annual chemical variability in the ice core.

We have updated Table A3 and edited the text in the methods (L231-233), results (L289-297) and discussion (L444-448) to reflect this.

Figure 4 and Discussion. A dot on each map for the ice core site would be helpful to orientate the reader. As you state, there is not a huge difference between the magnesium and sodium maps (because the records are very highly correlated, especially winter spring). Do you need to present both? At the moment its hard to interpret any differences, especially for winter spring as they may be artefacts of analysis more than anything.

Thank you for the suggestion. As there are some differences in summer, we decided to keep the text around magnesium but have moved the magnesium plots the appendix (Figure A4). We also added the location of the ISOL-ICE core to Figure 4.

Line 322 – what about orographic factors and relatively short sea salt aerosol life times of a few days? Can you prove transport across the continent, versus say short-term episodic inputs of high sea salt loads that can't be teased out in your 6 month means? I'm not suggesting you have to prove this, but I think the across continent statement is a big one and lacks evidence in this context, especially given the statement about air mass / accumulation source at lines 459-461. Would that mean that any cross-continental sea salt transport was dry deposited?

Of the two transport pathways, we agree that cross continental transport is less likely and have added text to support this.

L359-361 *"…and precipitation events take four days to arrive at Kohnen Station from the Southern Atlantic Ocean consistent with the relatively short sea salt aerosol lifetime of a few days in the Southern Ocean (Landwehr et al., 2022; Reijmer et al., 2002)."*

Minor comments:

In the abstract, I'd recommend separating into two sentences the sentence about the development of the age model, and then the snow accumulation and ice chemistry records and correlation. These are two separate (fairly major) steps and deserve a sentence each.

Line 27 – over the last two decades.

Done.

Line 66 – independently derived – do you mean via layer counting and volcanic horizons?

Yes. Text added L69 *"…from annual layer counting and volcanic horizons…"*

Line 279 – figure 4b? also, what is an 'offshore low' and an 'offshore high'?

We have referred to Figure 4b. Offshore refers to the low over the ocean as opposed to over the continent.

Line 369 – sea salt

Done.

Line 390 – what do you mean by a change in the DMS oxidation pathway

Text added L426 *"…change in DMS oxidation pathway from DMS to MSA and $SO_4^{2-}$…"*

**References**

Akers, P. D., Savarino, J., Caillon, N., Servettaz, A. P., Le Meur, E., Magand, O., Martins, J., Agosta, C., Crockford, P., and Kobayashi, K.: Sunlight-driven nitrate loss records Antarctic surface mass balance, Nature Communications, 13, 4274, 2022.

Cole-Dai, J. and Mosley-Thompson, E.: The Pinatubo eruption in South Pole snow and its potential value to ice-core paleovolcanic records, Annals of Glaciology, 29, 99-105, 1999.

Cole-Dai, J., Ferris, D. G., Lanciki, A. L., Savarino, J., Thiemens, M. H., and McConnell, J. R.: Two likely stratospheric volcanic eruptions in the 1450s CE found in a bipolar, subannually dated 800 year ice core record, Journal of Geophysical Research: Atmospheres, 118, 7459-7466, 2013.

Jiang, Z., Alexander, B., Savarino, J., and Geng, L.: An inverse model to correct for the effects of post-depositional processing on ice-core nitrate and its isotopes: model framework and applications at Summit, Greenland and Dome C, Antarctica, EGUsphere, 2023, 1-41, 10.5194/egusphere-2023-1054, 2023.

Landwehr, S., Volpi, M., Derkani, M. H., Nelli, F., Alberello, A., Toffoli, A., Gysel-Beer, M., Modini, R. L., and Schmale, J.: Sea state and boundary layer stability limit sea spray aerosol lifetime over the southern ocean, Authorea Preprints, 2022.

Langway, C., Osada, K., Clausen, H., Hammer, C., and Shoji, H.: A 10-century comparison of prominent bipolar volcanic events in ice cores, Journal of Geophysical Research: Atmospheres, 100, 16241-16247, 1995.

Plummer, C. T., Curran, M. A., van Ommen, T. D., Rasmussen, S. O., Moy, A. D., Vance, T. R., Clausen, H. B., Vinther, B. M., and Mayewski, P. A.: An independently dated 2000-yr volcanic record from Law Dome, East Antarctica, including a new perspective on the dating of the 1450s CE eruption of Kuwae, Vanuatu, Climate of the Past, 8, 1929-1940, 2012.

Reijmer, C., Van den Broeke, M., and Scheele, M.: Air parcel trajectories and snowfall related to five deep drilling locations in Antarctica based on the ERA-15 dataset, Journal of climate, 15, 1957-1968, 2002.

Sigl, M., McConnell, J. R., Layman, L., Maselli, O., McGwire, K., Pasteris, D., Dahl-Jensen, D., Steffensen, J. P., Vinther, B., and Edwards, R.: A new bipolar ice core record of volcanism from WAIS Divide and NEEM and implications for climate forcing of the last 2000 years, Journal of Geophysical Research: Atmospheres, 118, 1151-1169, 2013.

Traufetter, F., Oerter, H., Fischer, H., Weller, R., and Miller, H.: Spatio-temporal variability in volcanic sulphate deposition over the past 2 kyr in snow pits and firn cores from Amundsenisen, Antarctica, Journal of Glaciology, 50, 137-146, 2004.

Winton, V. H. L., Ming, A., Caillon, N., Hauge, L., Jones, A. E., Savarino, J., Yang, X., and Frey, M. M.: Deposition, recycling, and archival of nitrate stable isotopes between the air–snow interface: comparison between Dronning Maud Land and Dome C, Antarctica, Atmospheric Chemistry and Physics, 20, 5861-5885, 2020.

Zielinski, G. A., Mayewski, P. A., Meeker, L. D., Whitlow, S., Twickler, M. S., Morrison, M., Meese, D. A., Gow, A. J., and Alley, R. B.: Record of volcanism since 7000 BC from the GISP2 Greenland ice core and implications for the volcano-climate system, Science, 264, 948-952, 1994.

---

## Author Response (AR1)

Dear Prof. Hubertus Fischer

**RE: Authors' point-by-point response to cp-2023-96**

Thank you for the opportunity to revise our paper on the drivers of late Holocene ice core chemistry in Dronning Maud Land in accordance with the referees' comments. The referees provided thoughtful, helpful reviews which strengthened the paper. We believe the revised manuscript addresses the referee concerns and suggestions where possible. We greatly appreciate the feedback.

We outlined our point-by-point response to referee #1 and #2 in our online response posted in the interactive discussion.

Thank you for the additional wording suggestion which we have incorporated into the abstract in the revised manuscript. Below are the updated line numbers (marked up version) to help review the textural changes to referee #2 comments:

- *the issue of the accumulation range in DML to be suitable for the final goal of TCO reconstructions using d15NO3* (discussion L528-538 and L544-548).
- *an overview of how this paper fits into previous and future work done within ISOL-ICE* (introduction L87-99)
- *an improved analysis of the influence of ENSO on your* records (methods L234-236, results L292-300, discussion L447-451)

Best wishes
Holly Winton, Robert Mulvaney, Joel Savarino, Kyle Clem, and Markus Frey